



# Effect of Horizontal Resolution on the Simulation of Tropical Cyclones in the Chinese Academy of Sciences FGOALS-f3 Climate System Model

Jinxiao Li[1], Qing Bao[1], Yimin Liu[1], Lei Wang[1,5], Jing Yang[2,3], Guoxiong Wu[1], Xiaofei Wu[4], Bian He[1],
Xiaocong Wang[1], Xiaoqi Zhang[6,1], Yaoxian Yang[1], Zili Shen[7,1]

[1] State Key Laboratory of Numerical Modeling for Atmospheric Sciences and Geophysical Fluid Dynamics, Institute of
Atmospheric Physics, Chinese Academy of Sciences, Beijing 100029, China
[2] State Key Laboratory of Earth Surface Processes and Resource Ecology, Faculty of Geographical Science, Beijing Normal
University, Beijing 100875, China
[3] Southern Marine Science and Engineering Guangdong Laboratory, Guangzhou 511458, China
[4] School of Atmospheric Sciences/Plateau Atmosphere and Environment Key Laboratory of Sichuan Province, Chengdu
University of Information Technology, Chengdu 610225, China
[5] University of Chinese Academy of Sciences, Beijing 100049, China
[6] School of Atmospheric Sciences, Nanjing University of Information Science and Technology, Nanjing 210044, China
[7] Collaborative Innovation Center on Forecast and Evaluation of Meteorological Disasters, Nanjing University of Information
Science and Technology, Nanjing 210044, China

*Correspondence to*: Qing Bao (baoqing@mail.iap.ac.cn)

**Abstract.** The effects of horizontal resolution on the simulation of tropical cyclones were studied using the Chinese Academy of Sciences FGOALS-f3 climate system model from the High-Resolution Model Intercomparison Project (HighResMIP) for the Coupled Model Intercomparison Project Phase 6 (CMIP6). Both the low-resolution (approximately 100 km resolution) FGOALS-f3 model (FGOALS-f3-L) and the high-resolution (approximately 25 km resolution) FGOALS-f3 (FGOALS-f3-H) model were used to achieve the standard Tier1 experiment required by the HighResMIP. FGOALS-f3-L and FGOALS-f3-H have the same model parameterizations with the exactly the same parameters. The only differences between the two models are the horizontal resolution and the time step. The performance of FGOALS-f3-H and FGOALS-f3-L in simulating tropical cyclones was evaluated with the observation firstly. FGOALS-f3-H (25 km resolution) simulated more realistic distributions of the formation, movement and intensity of the climatology of tropical cyclones than FGOALS-f3-L at 100 km resolution. The seasonal cycles of the number of tropical cyclones increased by about 50% at the higher resolution and better matched the observed values in the peak month, especially in the eastern Pacific, northern Atlantic, southern Indian and southern Pacific oceans. The simulated variabilities of the number of tropical cyclones and the accumulated cyclone energy were both significantly improved from FGOALS-f3-L to FGOALS-f3-H over most of the ocean basins on the interannual timescale. The characteristics of the tropical cyclones (e.g., the average lifetime, the wind–pressure relationship and the horizontal structure) were more realistic in the simulation using the high-resolution model. The possible physical linkage between the performance of the tropical cyclone simulation and the horizontal resolution were revealed by further analyses. The improvement in the Madden–Julian oscillation from FGOALS-f3-H contributed to the realistic simulation of tropical cyclones. The genesis





potential index and the vorticity, relative humidity, maximum potential intensity and the wind shear terms were used to diagnose the effects of resolution. The current insufficiencies and future directions of improvement for the simulation of tropical cyclones and the potential applications of the FGOALS-f3-H model in the sub-seasonal to seasonal prediction of tropical cyclones are discussed.

## 1 Introduction

Tropical cyclones are extreme weather phenomena characterized by intense wind speeds and heavy rainfall. Although tropical cyclones alleviate coastal droughts, they can also cause severe economic losses and significant human casualties (Mendelsohn et al., 2012; Aon, 2018). Against the current background of global climate change, the effective simulation, prediction and projection of global tropical cyclone activity is challenging, but essential for disaster prevention and mitigation (Emanuel, 2017).

The simulation of tropical cyclones in global climate models (GCMs) is challenging in terms of both resolution and physical processes. Tropical cyclone-like structures appeared in early GCMs and Manabe et al. (1970), Bengtsson et al. (1982), Krishnamurti et al. (1989), Broccoli and Manabe (1990), Wu and Lau (1992) and Haarsma et al. (1993) were pioneers in using objective feature-tracking approaches to study simulated tropical cyclones. However, the low resolution and incomplete parameterization of the physical processes in these early GCMs meant that their performance in simulating tropical cyclones

was limited. For this reason, statistical methods were used to study on the climatology of tropical cyclones. Camargo et al. (2013) found that the simulation of the frequency of tropical cyclones in the Coupled Model Intercomparison Project 5 (CMIP5) was much lower than in the observations. This was mainly due to the cold biases of the sea surface temperature, which amplified the uncertainty of future projections. Emanuel (2013) designed a downscaling method to reduce the uncertainty in projections of tropical cyclone activity.

The horizontal and vertical resolutions of climate system models increased over the following half-century in line with the complex parameterization of the physical processes. As a result, more refined details (e.g., tropical cyclones and tropical waves) can now be resolved. Regional climate system models with smaller spatial scales and lower computing costs can now be used to simulate tropical cyclones. Knutson et al. (2007) used a high-resolution regional model to simulate tropical cyclone activity in the northern Atlantic Ocean. The structure and interannual variability of the simulated tropical cyclones has a high

fidelity with the observations. There has also been a significant increase in the resolution of GCMs. Oouchi et al. (2016) used a 20 km mesh global atmospheric model to simulate tropical cyclone activity in a warming climate and found that the high-resolution GCM could not only describe the details of typhoons very well, but also captured the variability of tropical cyclones. Zhao et al. (2009), Knutson et al. (2010), Murakami et al. (2012), Manganello et al. (2012), Strachan et al. (2013) and Zarzycki et al. (2015) showed that high-resolution GCMs can simulate many of the characteristics of tropical cyclones. The uncertainty

in simulating tropical cyclones has been reduced with an increase in resolution, the reasonable parameterization of the physical processes and improvements in the downscaling method (Walsh et al., 2015, 2016; Camargo et al., 2016).




The increase in the horizontal resolution of GCMs has led to significant changes in the simulation of the variability of tropical cyclones. Previous studies showed that there are significant changes in the El Niño–Southern Oscillation (ENSO) as the horizontal resolution of GCMs increases (Philander et al., 1992; Kuntson et al., 1997; Schneider et al., 2003; Masson et al.,

2012; Larson et al., 2013; Meehl et al., 2020) and the simulation results were mostly positive. However, these improvements in predicting the ENSO with an increase in horizontal resolution did not lead to improvements in the relationship between the ENSO and tropical cyclones (Matsuura et al., 1999; Bell et al., 2014; Krishnamurthy et al., 2016). There is also a close relationship between the Madden–Julian oscillation (MJO) and tropical cyclones (Liebmann et al., 1994; Hall et al., 2001; Camargo et al., 2008). High-resolution GCMs need to not only give a better description of the structure of tropical cyclones,

but should also well simulate the relationship between tropical cyclones and large-scale variabilities (e.g., the MJO and ENSO), which is crucial in reducing the uncertainties in the simulation and prediction of tropical cyclones (Manganello et al., 2012; 2016; Zhang et al., 2016; Delworth et al., 2020). As the horizontal resolution increases in the models, some key parameters in the physical parameterizations are tuned to give a better performance (Bacmeister et al., 2013; Roberts et al., 2020), e.g., Lim et al. (2015) found that an increase in the threshold of minimum entrainment led to the increasing TC activity, and Murakami

et al. (2012) found that the constrained convective heating in the convective scheme induced intense grid-scale upward motions and promoted large-scale condensation, which was favorable for the development of a more intense TC. These artificial tuning might introduce more uncertainties in terms of the effects of resolution, giving rise to conclusions that are controversial to the tropical cyclone research community.

The impacts of horizontal resolution on the simulation of tropical cyclones were studied using the Chinese Academy of

Sciences Flexible Global Ocean–Atmosphere–Land System, Finite-Volume Version 3 (FGOALS-f3) model, which was developed by the State Key Laboratory of Numerical Modeling for Atmospheric Sciences and Geophysical Fluid Dynamics (LASG), Institute of Atmospheric Physics (IAP). The simulated tropical cyclones in FGOALS-f3 were introduced firstly, then the outputs of FGOALS-f3-L and FGOALS-f3-H were used to reveal the influence of horizontal resolution on these simulations. The latest version of FGOALS-f3 participated in the CMIP6 (Eyring et al. 2016), DECK and MIPs endorsements

(Zhou et al. 2016; He et al. 2019, 2020; Haarsma et al. 2016). FAMIL2 is the atmospheric component of the climate system model FGOALS-f3. Li et al. (2019) evaluated the simulation performance of tropical cyclone activity in the latest generation atmospheric general circulation models from the LASG–IAP (FAMIL2) using a coarse resolution with standard AMIP experiments. Although FAMIL2 is able to reproduce the many aspects of the activities of tropical cyclones with a horizontal resolution of 1°, there is still some room for improvement in simulating tropical cyclones, such as the weak intensity of tropical

cyclones, fewer tropical cyclones in the peak month in the northern Atlantic and eastern Pacific oceans and inaccurate large-scale factors. Therefore, the HighResMIP configuration has been applied for both the low- and high-resolution FGOALS-f3. Both model versions retained the exact model physics and parameters and the only differences were the horizontal resolutions and model time steps, which better meet the rule of HighResMIP: "The experimental set-up and design of the standard resolution experiments will be exactly the same as for the high-resolution runs". This study aimed to address the following





issues: (1) the impacts of horizontal resolutions on the simulation of global tropical cyclones in a climate system model; and (2) the possible physical linkages between the horizontal resolutions and the simulated tropical cyclones?

This paper is organized as follows. Section2 introduces the model, data and methods used in this study. Section 3 shows the performances of simulated tropical cyclones in both FGOALS-f3-L and FGOALS-f3-H. Section 4 discusses the possible reasons for the improvement of the simulation of tropical cyclones with increased horizontal resolutions. Section 5 introduces

the physical parameterization and its impact on the simulated TC in GCMs, then discusses the potential value-added effect of MJO and TC due to the increase in horizontal resolution from HighResMIP models. Section 6 provides a summary of the results.

## 2 Model, data and methods

### 2.1 Description of FGOALS-f3

FGOALS-f3 is the latest version of the Chinese Academy of Sciences climate system model and was designed for CMIP6. The FGOALS-f3 model consists of four components: (1) the atmospheric component is the Finite-volume Atmospheric Model Version 2.2 (FAMIL2.2) (Zhou et al., 2015; Bao et al., 2018, 2020; Li et al., 2019; He et al., 2019), which is the successor to the atmospheric general circulation model of the Spectral Atmosphere Model of LASG (SAMIL) (Wu et al., 1996; Bao et al., 2010, 2013); (2) the oceanic component is the LASG/IAP Climate System Ocean Model Version 3 (LICOM3) (Liu et al.,

2012); (3) the land surface component is the Community Land Model Version 4.0 (CLM4) (Oleson et al., 2010; Lawrence et al. 2011), with the processes from the dynamic global vegetation model in CLM4.0 used in FGOALS-f3-L/H turned off; and (4) the sea ice component is the Los Alamos Sea Ice Model Version 4.0 (CICE 4.0; Hunke et al. 2008; Hunke & Lipscomb, 2010). These four components are coupled by the Version 7 coupler in the CESM (Craig et al., 2012). Li et al. (2019) introduced the atmospheric component FAMIL2 in detail and carried out some tuning to achieve stability in long-term coupled

integrations (defined as FAMIL2.2).

As is shown in Table 1, the finite-volume cubed-sphere dynamical core (FV3) (Lin, 2004; Putman and Lin, 2007; Voosen, 2017) is used as the dynamical core in FAMIL2.2, which is the atmospheric component of FGOALS-f3. The University of Washington moist turbulence parameterization (Park & Bretherton, 2009) is also used in FGOALS-f3. This is a non-local, high-order closure scheme and uses the diagnosed turbulent kinetic energy to determine the eddy diffusivity in turbulence. The

Resolving Convective Precipitation parameterization (Bao and Li, 2020) is used, which involves calculating the microphysical processes in the cumulus scheme for both deep and shallow convection; six species are considered, similar to the Geophysical Fluid Dynamics Laboratory (GFDL) cloud microphysics scheme (Zhou et al., 2019). The gravity wave drag scheme (Palmer et al., 1986), the cloud fraction diagnosis scheme (Xu & Randall, 1996) and the radiative transmission scheme (Clough et al., 2005) are also considered.

The vertical layers of FGOALS-f3-L and FGOALS-f3-H are both set to 32, whereas the horizontal resolutions of FGOALS-f3-L and FGOALS-f3-H are C96 (approximately 100 km) and C384 (approximately 25 km), respectively (Table 2).



To maintain the stability of the integration for the dynamical core, the two parameters k_split and n_split included in FV3 are different in FGOALS-f3-L and FGOALS-f3-H. k_split is the number of vertical remapping operations per physical time step in the dynamical integration and n_split is the number small dynamic (acoustic) time steps between the vertical remapping

operations, which will affect the stability of the integration when the horizontal resolution of the model is changed. Considering that FGOALS-f3-H requires more frequent vertical remapping, k_split and n_split are set to 6 and 15, respectively (they are 2 and 6, respectively, in FGOALS-f3-L). The time steps of the physical processes are both set to 30 minutes, but the update frequency of radiative transmission and the minimum time step of the microphysics scheme in both FGOALS-f3-L and FGOALS-f3-H are 1 h and 150 s, respectively. Li et al. (2017) tested the computing performance between the FGOALS-f3-L

and FGOALS-f3-H using the Supercomputer Tianhe-2 and the results indicated a high computing speed-up and low computing costs when the number of parallel processes was increased.

**2.2 Datasets**

FGOALS-f3 were participated in CMIP6 DECK (Eyring et al., 2016), the Global Monsoons Model Inter-comparison Project (GMMIP; Zhou et al. 2016; He et al. 2019) and the High-Resolution Model Intercomparison Project (HighResMIP;

Haarsma et al. 2016). The datasets from CMOR for the CMIP6 HighResMIP are in a standard format. The experimental design satisfies the requirement for HighResMIP. Two resolutions of FGOALS-f3 are used to compare the simulation of tropical cyclone activities at different resolutions. To identify the impact of resolution, there is no tuning of the parameterization of the physical processes between FGOALS-f3-L and FGOALS-f3-H. The time period 1991–2014 is extracted to avoid the uncertainties in the pre-satellite era in the observations.

The International Best Track Archive for Climate Stewardship v03r10 (IBTrACS; Knapp et al., 2010) is used as the observational dataset. IBTrACS is a multisource dataset and includes the RSMC Tokyo, Chinese Meteorological Administration–Shanghai Typhoon Institute (Ying et al. 2014) and the National Hurricane Center data sources. To provide a fair comparison, transformation from the 10-min-averaged maximum sustained wind to the 1-min-averaged maximum sustained wind is needed and the relevant coefficient is set to 0.88 (Manganello et al., 2012; Knapp et al. 2010; Li et al. 2019).

The European Centre for Medium-Range Weather Forecasts Re-Analysis ERA-Interim dataset (Dee et al., 2011) (resolution 0.75°), the National Centers for Environmental Prediction Global Forecast System (GFS) reanalysis dataset (resolution 0.25°) and the Global Precipitation Measurement (GPM) dataset (resolution 0.25°) (Hou et al., 2014) in the time period 1991–2014 are used as the observations to quantitatively evaluate simulated tropical cyclones in FGOALS-f3.

**2.2 Tracking algorithms**

An objective feature-tracking approach is used to detect the model-generated tropical cyclones based on the 6-h outputs of FGOALS-f3-L and FGOALS-f3-H. According to the tracking scheme (Table 3), the sea-level pressure, warm core (the temperature anomaly averaged between 300 and 500 hPa), 10 m wind and the 850 hPa absolute vorticity are used to diagnose the tropical cyclone activity, which is similar to the method used in the climate system model of the GFDL (Zhao et al. 2009;



Chen & Lin 2013; Xiang et al. 2015). Li et al. (2019) used this scheme to evaluate the simulated performance of tropical
cyclones in FAMIL2 and showed a consistent performance. The wind speed thresholds between FGOALS-f3-L and FGOALS-
f3-H are consistent with the relationship between the horizontal resolution of the models and the tropical cyclone detection
algorithms (Walsh et al., 2007).

## 3 Results

### 3.1 Global climatology of tropical cyclone track density

The climatology of simulated tropical cyclones is the first step in testing the performance of the model. Zhao et al. (2009)
used a GFDL GCM with a 50 km horizontal resolution to simulate global tropical cyclone activity and obtained a negative
bias in the number of tropical cyclones in the eastern Pacific, northern Atlantic and southern Indian oceans. These biases also
appeared in the low-resolution models participating in the US CLIVAR Working Group on Hurricanes (Walsh et al., 2015).
Figure 1a and 1b show the tracks and intensities of global tropical cyclones in the FGOALS-f3-L and FGOALS-f3-H
simulations, respectively. The definition of the intensity follows the Saffir–Simpson intensity scale (Simpson & Saffir, 1974).
The negative biases of the tracks of tropical cyclones in FGOALS-f3-L (Figure 1a) improves when the horizontal resolution
increases from 100 to 25 km (Figure 1b) relative to IBTrACS. This improvement is clearly shown in Figure 2 and the
differences in track densities between FGOALS-f3-H and FGOALS-f3-L reflect a positive distribution in the global basin
(Figure 2c). The biases of the track densities between the simulation and IBTrACS are improved when the horizontal resolution
is increased from 100 km (Figure 2a) to 25 km (Figure 2b), especially in the low-latitude basins where tropical cyclones form.
Negative biases between FGOALS-f3-L and IBTrACS appear in the mid-latitudes of the western Pacific and northern Atlantic
oceans, but positive biases between the FGOALS-f3-H and IBTrACS also appear in these areas, which means that there are
more tropical cyclone events at higher latitudes in FGOALS-f3-H than in IBTrACS and the simulation in FGOALS-f3-L. This
phenomenon also exists in the high-resolution GCMs that participated in the European Union Horizon 2020 project
PRIMAVERA (Roberts et al., 2020).

The negative biases of the intensities of tropical cyclones are also improved when the horizontal resolution is increased
from 100 km (Figure 1a) to 25 km (Figure 1b), especially in the western Pacific and northern Atlantic oceans. The difference
in wind speed densities between FGOALS-f3-L and FGOALS-f3-H (Figure 3c) shows a significant increase in the wind speed
densities in the northern Atlantic, eastern Pacific and northern Indian oceans and the mid-latitude region of the western Pacific
Ocean when the horizontal resolution is increased from 100 to 25 km. Similar to the pattern of track density anomalies (Figure
2), the biases of the wind speed densities between the FGOALS-f3 and IBTrACS are improved when the horizontal resolution
is increased from 100 to 25 km, but the positive biases are intensified in the mid-latitudes of the western Pacific and northern
Atlantic oceans. Figure 4 shows the pressure–wind pairs for each 6-hourly measurement of tropical cyclones between
FGOALS-f3 and IBTrACS. These results indicate that the spread of pressure–wind pairs in FGOALS-f3-L is narrow and there
is a severe underestimation of intense tropical cyclone events at the lower surface pressures and higher wind speeds in the


western Pacific and northern Atlantic oceans (Figure 4a, 4c), although this bias has been dramatically improved (Figure 4b, 4d).

The increased intensity of tropical cyclones in FGOALS-f3-H favors the apparent negative bias of the tropical cyclone lifetime in FGOALS-f3-L when the horizontal resolution is increased. Figure 5 shows the average lifetime of tropical cyclones

from 1991 to 2014. The average lifetime of tropical cyclones in the observations is about 8.5, 7.5, 7.5, 4 and 7.5 days in the western Pacific, northern Atlantic, eastern Pacific, northern Indian and southern Pacific oceans, respectively, and the simulation of the average lifetime of tropical cyclones is increased in these five basins when the horizontal resolution is increased from 100 to 25 km. For example, the simulated average lifetime of tropical cyclones increases from 6 to 7.4 days when the horizontal resolution is increased in the western Pacific Ocean.

**3.2 Seasonal cycles and the interannual variability of tropical cyclones**

Evaluating the seasonal cycle and interannual variability of tropical cyclones in GCMs is an efficient way to verify the coordination between tropical cyclone activity and large-scale circulation patterns (Manganello et al., 2012; Camargo et al., 2016; Kuntson et al., 2019; 2020). Robert et al. (2020) found no uniform improvement in the seasonal cycle and interannual variability of tropical cyclones at increased horizontal resolutions, which means that there is a difference in coordination

between tropical cyclone activity and large-scale circulation patterns in high-resolution GCMs. Figure 6 shows the seasonal cycle of tropical cyclones between IBTrACS, FGOALS-f3-L and FGOALS-f3-H and shows that FGOALS-f3-L gives a consistent underestimation of the seasonal cycle of tropical cyclones in the northern Atlantic, eastern Pacific, northern Indian and southern Pacific oceans. Neither the single peak in the number of tropical cyclones in the northern Atlantic (peak month September), eastern Pacific (peak month August) and southern Pacific (peak month February) oceans nor the double peak in

the northern Indian Ocean (peak months May and November) could be reproduced in FGOALS-f3-L. There are two increases in the simulated number of tropical cyclones in the peak months of the northern Atlantic and eastern Pacific oceans and the characteristics of the seasonal cycle of simulated tropical cyclones are improved in the northern Indian, western Pacific and southern Pacific oceans as the horizontal resolution is increased from 100 to 25 km. Figure 7 shows the interannual correlation of the numbers of tropical cyclones between FGOALS-f3 and IBTrACS. The results show that the correlation coefficient is

improved in each basin, which reflects the fact that the interannual variability between the tropical cyclone activity and large-scale circulation patterns is harmonious.

Figure 8 shows the interannual correlation of the accumulated cyclone energy (ACE; Bell et al., 2000) between FGOALS-f3 and IBTrACS. ACE is a measure used by the National Oceanic and Atmospheric Administration, which means that the energy over the lifetime of a tropical cyclone is calculated for every 6-h period:

$$ACE = 10^{-4} \sum V_m^2 \tag{1}$$

where $V_m$ is the estimated sustained wind speed in knots. The results (Figure 8) show that the correlation coefficient of the ACE between IBTrACS and the simulation is improved in each basin when the horizontal resolution is increased from 100 to





25 km. The increase in the number, lifetime and intensity of simulated tropical cyclones contributes to the increased correlation coefficient of ACE.

## 3.3 Horizontal structure of tropical cyclones

Previous studies have shown that the horizontal resolution influences the horizontal structure of simulated tropical cyclones (Strachan et al., 2013; Murakami et al., 2012; 2013; Roberts et al., 2020). Manganello et al. (2012) compared the horizontal structure of moisture content of TC of GCM between low (T511) and high (T2047) resolutions and found that the refined structure of tropical cyclone liquid was simulated when the horizontal resolution is increased. Following the method of Manganello et al. (2012), the surface 10 m wind and daily precipitation rate are combined, which are from the 30 most intense tropical cyclones between FGOALS-f3-L (Figure 9a) and FGOALS-f3-H (Figure 9b). As a comparison, the results of the GFS (Figure 9c) and the GFS&GPM (Figure 9d) simulations are also shown. The results indicate that there is no prominent precipitation structure of tropical cyclones in FGOALS-f3-L, but there is a considerable improvement in the results with a horizontal resolution of 25 km. The eyewall and organized precipitation of tropical cyclones are apparent in the FGOALS-f3-H simulation. The main uncertainty is the extreme position of precipitation. The extreme position of precipitation appears 100 km east of the eyewall in the FGOALS-f3-H simulation (Figure 9b), but is 100 km south of the eyewall in the GFS&GPM datasets (Figure 9c).

## 4 Possible reasons for the improvement in the simulation of tropical cyclones at increased horizontal resolutions

## 4.1 Modulation of the tropical cyclone track by the MJO

There is a clear evidence of the connection between the MJO and tropical cyclone activity worldwide (Camargo et al., 2009; Klotzbach et al., 2014). Zhang et al. (2013) summarized the connection between the MJO and global tropical cyclone activity and found that the MJO affects the formation and movement of tropical cyclones in each phase. Figure 10 shows the composite May–October 20–100-day precipitation as a function of phase of the MJO between FGOALS-f3-L (Figure 10a, c, e, g, i, k, m and o) and the observations (Figure10b, d, f, h, j, l, n and p). One of the major biases in the precipitation modulated by the MJO relative to the observations appears in the stage during which it spreads from the Indian Ocean to the eastern Pacific Ocean and especially in phases 4–6, which is the wet phase (positive precipitation anomalies of the MJO) in the western Pacific, eastern Pacific and northern Atlantic oceans. Negative precipitation anomalies are seen in FGOALS-f3-L from phases 4 to 6 (Figure 10g, i and k) when positive precipitation anomalies are seen in the observation (Figure 10h, j and l) in the western Pacific, eastern Pacific and northern Atlantic oceans. These negative precipitation anomalies of MJO inhibit the formation and propagation of tropical cyclones in FGOALS-f3-L. A considerable improvement in the simulated propagation of the MJO in the northern Indian, western Pacific, eastern Pacific and northern Atlantic oceans occurs when the horizontal resolution is increased (Figure 11) and this provides an accurate background for the formation and propagation of tropical cyclones.





Using the phase of the MJO to produce a composite of the daily track densities of tropical cyclones is an effective way of exploring the coordination between the MJO and tropical cyclone activity in GCMs. The biases of track density anomalies (Figure 12) are in agreement with the precipitation anomalies in the MJO (Figure 10). The results indicate that FGOALS-f3-L cannot reasonably reproduce the pattern of track densities in each phase of the MJO in the northern Indian, western Pacific, eastern Pacific and northern Atlantic oceans. There is considerable improvement in the track densities modulated by the MJO when the horizontal resolution is increased (Figure 13). The improvement in the MJO with an increase in the horizontal resolution plays a crucial role in simulating the variability of tropical cyclones at sub-seasonal to seasonal scales.

## 4.2 Large-scale environmental factors

The genesis potential index (GPI; Emanuel et al., 2004) is applied to detect the connection between the genesis of tropical cyclones and large-scale circulation patterns. Camargo et al. (2007) and Walsh et al. (2013) found that the correlation between the GPI and the variation of tropical cyclones in GCMs mainly depends on the horizontal resolution and the similarity between the GPI. The variation in tropical cyclones is increased when the horizontal resolution is increased. The GPI used in this work is defined as:

$$GPI = |10^5 vort850|^{3/2} \left(\frac{RH}{50}\right)\left(\frac{V_m}{70}\right)(1 + 0.1V_{shear})^{-2} \tag{2}$$

where $vort850$ is the 850 hPa absolute vorticity (s$^{-1}$), $RH$ is the 600 hPa relative humidity (%), $V_m$ is the maximum potential intensity (Emanuel, 1995) and $V_{shear}$ is the magnitude of the wind shear between 850 and 200 hPa (m s$^{-1}$). $V_m$ (the maximum potential intensity) is defined here as:

$$V_m = \frac{C_k T_s}{C_d T_0}\left(CAPE^* - CAPE^b\right) \tag{3}$$

where $C_k$ is the exchange coefficient of the enthalpy, $C_d$ is the drag coefficient, $T_s$ is the sea surface temperature and $T_0$ is the mean outflow temperature. $CAPE^*$ is the convective available potential energy of the air lifted from saturation at sea-level and $CAPE^b$ is the convective available potential energy of the boundary layer air.

Figure 14 shows the GPI for the FGOALS-f3-L (Figure 14a), FGOALS-f3-H (Figure 14b) and ERAI (Figure 14c) simulations. There is a considerable underestimation of the GPI in FGOALS-f3-L, which is more remarkable in the eastern Pacific and northern Atlantic oceans. These biases are consistent with the biases of the simulated tropical cyclones (Figure 2a). There is more consistency between the GPI in the observations (Figure 14c) and the simulated frequency of tropical cyclones (Figure 14b) when the horizontal resolution is increased. The GPI equation is split into four parts (Figure 15) and calculated the pattern correlation of each individual part. P1 represents the equation $|10^5 vort850|^{3/2}$, P2 represents the equation $\frac{RH}{50}$, P3 represents the equation: $\frac{V_m}{70}$ and P4 represents the equation $(1 + 0.1V_{shear})^{-2}$. The results indicate that the pattern correlation coefficients are increased in the northern Indian (Figure 15a), western Pacific (Figure 15b), eastern Pacific ((Figure 15c) and northern Atlantic (Figure 15d) oceans, which is favored by the reduction in the bias of the large-scale circulation patterns.





## 5 Discussions

### 5.1 The physical parameterization and its impact on the simulated TC in GCMs

A Resolving Convective Precipitation (RCP) scheme has been used in both the high and low versions of FGOALS-f3 (Bao and Li, 2020; He et al., 2019; Li et al., 2019). The RCP scheme calculates convective and stratiform precipitation at the grid scale, which has the advantage of both scale-awareness and high computational efficiency. The parameterizations of physical processes in traditional GCMs are very sensitive to the change of resolution. Especially, the processes of convection and clouds is considered as effective resolved, which means the assumptions and equations in the low-resolution condition are not suitable for the high-resolution GCMs. As the result, the model convergence with increasing resolution will be degraded (Sakradzija et al., 2016). Simulated TC are very sensitive to the processes of convection and cloud in GCMs (Zhao et al., 2012). Actually, the effective tunning for the convection (Lim et al., 2015; Murakami et al., 2012), boundary (Zhang et al., 2017), and microphysics (Chutia et al., 2019) parameterizations will contribute to the improvement of intensity, number, track, and structure of simulated TC. Although the fixed parameterization scheme combined with the fine grid will improve the simulation performance of TC obviously, the effect of resolution will be amplified. According to this study, the GCM with the scale-aware parameterizations still slightly underestimates the intensity of TC at 0.25° degree resolution (Figure 1). Besides, air-sea exchanges and non-hydrostatic processes are both important to enhance intense of TC (Ma et al., 2017). Emanuel and Sobel, (2013) found that the absence of air-sea coupling can lead to potentially large imbalances in the surface energy budget, which is not conductive to the development of TC.

### 5.2 The potential value-added effect of MJO and TC due to the increase in horizontal resolution from HighResMIP models

The results show a clear improvement in the relationship between the MJO and tropical cyclone activity when the horizontal resolution of FGOALS-f3 is increased from 100 to 25 km. This result indicate that the large-scale background associated to the TC is improved when increased the horizontal resolution of FGOALS-f3. It is worth exploring whether this improved relationship is common to all the GCMs participating in CMIP6. The eight models participating in the HighResMIP Tier1 were selected (Table 4) to calculate the MJO phases and GPI separately. Figure 16 shows the ranking of the models according to the average anomaly correlation coefficient of the MJO from phases 1 to 8 and shows that there is an increase in the anomaly correlation coefficient of the MJO in the high-resolution models (red cylinders) relative to the low-resolution models (blue cylinders). The MJO phases simulated by the models could provide an accurate large-scale background for the generation and development of tropical cyclones (Zhang et al., 2013). Figure 17 shows the anomaly of the composite GPI by MJO phases 4–7 between the multi-model mean of the GCMs and the ERA-Interim dataset and the result indicates that the biases of the GPI in the South China Sea and the western north Pacific Ocean are decreased in the high-resolution models (Figure 17b) relative to the low-resolution models (Figure 17a) as a result of the improved simulation of the MJO phases when





the horizontal resolution of the models is increased. Although there seems to be a significant improvement in the MJO-TC relationship when the horizontal resolutions of the GCMs are increased, it is worth noting that not all the GCMs participate in the HighResMIP follow the rules: "The experimental set-up and design of the standard resolution experiments will be exactly the same as for the high-resolution runs", which mean the specific optimization in the parameterization of physical processes
for the GCMs in the high horizontal resolution (Roberts et al., 2020). The consequence is that the effect of horizontal resolution on TC simulation will be overestimated or underestimated (Strachan et al., 2013).

**6 Summary and conclusions**

The impacts of horizontal resolution on the simulation of tropical cyclones were studied with the latest version of FGOALS-f3, which participated in CMIP6 HighResMIP (Haarsma et al., 2016). Li et al. (2019) evaluated the simulation
performance of tropical cyclone activity in FAMIL2 (resolution about 100 km), which is the atmospheric component of FGOALS-f3 (He et al., 2019), and put forward the idea that the simulated performance of tropical cyclones is improved with the increased horizontal resolution in FAMIL2. These hypotheses were examined in this study and the  main findings and conclusions are as follows.

1. There are improvements in the track and intensity of global tropical cyclones in the simulation of FGOALS-f3 when
the horizontal resolution is increased from 100 (FGOALS-f3-L) to 25 km (FGOALS-f3-H) and the negative biases in tropical cyclone genesis are improved in the eastern Pacific and northern Atlantic oceans. Quantitative comparisons between the track density of tropical cyclones between FGOALS-f3-L, FGOALS-f3-H and IBTrACS show that the negative biases of the tropical cyclone track densities at low latitudes are alleviated. The surface wind speed of tropical cyclones is increased when the horizontal resolution is increased and this change in FGOALS-f3-H is closer to the
observations. The improvement in the intensity of tropical cyclones in FGOALS-f3-H is easier to detect in the pressure–wind pairs of tropical cyclones. A wide spread of pressure–wind pairs is simulated in FGOALS-f3-H and the biases in the pressure–wind pairs are improved relative to the observations.

2. The global lifetimes of tropical cyclones in FGOALS-f3-H are increased compared with the FGOALS-f3-L simulation, especially in the western Pacific, northern Atlantic, eastern Pacific and southern Pacific oceans. The increase in the
345 intensity of tropical cyclones in FGOALS-f3-H contributes to the improvement in the lifetime of tropical cyclones. In the seasonal cycle of tropical cyclones, only 50% of tropical cyclones are simulated in the peak month, but this bias is improved in FGOALS-f3-H compared with the observations. The correlation of the annual number of tropical cyclones and the annual ACE of tropical cyclones are both improved in the western Pacific, northern Atlantic, eastern Pacific, southern Pacific and northern Indian oceans when the horizontal resolution is increased.

3. There is a significant improvement in the horizontal structure of tropical cyclones in FGOALS-f3 when the horizontal resolution is increased. The eyewall and the organized precipitation of tropical cyclones are apparent in the FGOALS-f3-H simulation and the main uncertainty is the extreme position of precipitation.



4. The possible reasons for the improvement in the simulated tropical cyclones are physically explained by the MJO and large-scale environmental factors. FGOALS-f3-H can reproduce the propagation of MJO realistically, which provides a favorite background for the formation and propagation of tropical cyclones on sub-seasonal to seasonal scales. The GPI biases between FGOALS-f3-L and FGOALS-f3-H are well consistent with the biases in the simulated tropical cyclones. The improvement in the large-scale environmental factors in FGOALS-f3-H contributes directly to the simulation of tropical cyclones. This study shows that it is worth establishing a high-resolution coupled dynamic prediction system based on FGOALS-f3-H to improve the prediction skill of tropical cyclones on sub-seasonal to seasonal scales (Camp et al., 2018; Murakami et al., 2016). This dataset will be uploaded to the sub-seasonal to seasonal prediction project (Vitart, et al., 2018; Vitart et al., 2017).

**Code and data availability**

The model output of FGOALS-f3 models for CMIP6 simulations, which is used in this work is uploaded to the Earth System Grid Fedration (ESGF), and the users can access to these outputs freely. The DOI for 'CMIP6.HighResMIP.CAS.FGOALS-f3-H. highresSST-present' is doi:10.22033/ESGF/CMIP6.3312, and the DOI for 'CMIP6.HighResMIP.CAS.FGOALS-f3-L. highresSST-present' is doi:10.22033/ESGF/CMIP6.12009. The other output of GCMs participate in HighResMIP (Table 4) are also distributed through ESGF, and the users can search it on the website: https://esgf-node.llnl.gov/projects/cmip6/ after a simple registration. The source code of the model can be found in DOI: http://doi.org/10.5281/zenodo.4588109. All the source code and data can also be available on request to the corresponding author Qing Bao (baoqing@mail.iap.ac.cn).

**Author contributions**

Qing Bao led the CAS FGOALS-f3 development, and all other co-authors contributed to it. Jinxiao Li participated in the development of CAS FGOALS-f3 and evaluated the simulation performance of tropical cyclone in FGOALS-f3. Yimin Liu, Lei Wang, Jing Yang, Guoxiong Wu, Xiaofei Wu, Bian He, Xiaocong Wang, Xiaoqi Zhang, Yaoxian Yang, and Zili Shen contributed to the manuscript writing for this work.

**Competing interests**

The authors declare that they have no conflict of interest.

**Acknowledgements**

This work was jointly funded by the Strategic Priority Research Program of Chinese Academy of Sciences [grant numberXDB40030205], the National Natural Science Foundation of China [grant numbers 42005117, 91737306, 41675100, and U1811464], and the Key Special Project for Introduced Talents Team of Southern Marine Science and Engineering Guangdong Laboratory (Guangdong) [grant numbers GML2019ZD0601].



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





**Table 1.** Dynamical core and physics parameterization schemes used in FGOALS-f3.

| Model configuration | FGOALS-f3 |
| --- | --- |
| Dynamical core | Finite-Volume Cubed-Sphere Dynamical core |
| Boundary layer scheme | University of Washington Moist Turbulence parameterization (Park & Bretherton, 2009) |
| Radiation scheme | Rapid Radiative Transfer Model for GCMs (Clough et al., 2005) |
| Gravity wave drag scheme | Palmer et al. (1996) |
| Cloud fraction diagnosis scheme | Xu & Randall (1996) |
| Convection microphysics scheme | Resolving Convective Precipitation (Bao and Li, 2020) |





**Table 2.** Comparison of resolutions and time steps in FGOALS-f3.

| Model configuration | 100 km FGOALS-f3 | 25 km FGOALS-f3 |
|---|---|---|
| Horizontal resolution | C96 (about 100 km) | C384 (about 25 km) |
| Number of vertical layers | 32 layers | 32 layers |
| Number of vertical remapping operations per physical time step with dynamical integration (k_split) | 2 | 6 |
| Number of small dynamic time steps between the vertical remapping operations (n_split) | 6 | 15 |
| Time step of dynamical core | 30 min | 30 min |
| Time step of physical processes | 30 min | 30 min |
| Frequency of radiative transmission | 1 h | 1 h |
| Minimum time step of microphysics scheme | 150 s | 150 s |





**Table 3.** Comparison of tropical cyclone identification criterion between FGOALS-f3-L and FGOALS-f3-H.

| Variable | FGOALS-f3-L | FGOALS-f3-H |
|---|---|---|
| Surface wind speed threshold (m s$^{-1}$)* | ≥14.0 | ≥15.4 |
| 850 hPa absolute vorticity (s$^{-1}$) | ≥3.5 × 10$^{-5}$ | ≥3.5 × 10$^{-5}$ |
| Warm core (average temperature between 300 and 500 hPa; ℃) | ≥1 | ≥1 |
| Lifetime (h) | ≥72 | ≥72 |

\* The criteria for the surface wind speed were corrected between FGOALS-f3-L and FGOALS-f3-H (Walsh et al., 2007).



**Table 4.** The information of the GCMs that participate in HighResMIP.

| Institution | Model name | Horizontal resolution/ nominal resolution | References |
|---|---|---|---|
| Chinese Academy of Sciences (CAS) | FGOALS-f3-L | C96/100km | Bao et al., 2020 |
| | FGOALS-f3-H | C384/25km | He et al., 2019 |
| | | | Li et al., 2019 |
| Meteorological Research Institute (MRI) | MRI-AGCM3-2-H | T319/50km | Mizuta et al., 2012 |
| | MRI-AGCM3-2-S | T959/25km | Murakami et al., 2012 |
| Max Planck Institute for Meteorology (MPI) | MPI-ESM1-2-HR | T127/100km | Müller et al., 2018 |
| | MPI-ESM1-2-XR | T255/50km | Gutjahr et al., 2019 |
| Institute Pierre Simon Laplace (IPSL) | IPSL-CM6A-LR | N96/250km | Boucher et al., 2020 |
| | IPSL-CM6A-ATM-HR | N256/50km | |
| European Centre for Medium-Range Weather Forecasts (ECMWF) | ECMWF-IFS-LR | Tco199/100km | Roberts et al., 2018 |
| | ECMWF-IFS-HR | Tco399/25km | |
| EC-Earth-Consortium | EC-Earth3P | T255/100km | Haarsma et al., 2020 |
| | EC-Earth3-HR | T511/50km | |
| Centre National de Recherches Meteorologiques (CNRM) | CNRM-CM6-1 | T127/250km | Voldoire et al., 2019 |
| | CNRM-CM6-1-HR | T359/50km | |
| Met Office Hadley Centre | HadGEM3-GC31-LM | N96/250km | Williams, K., et al |
| | HadGEM3-GC31-HM | N512/50km | |






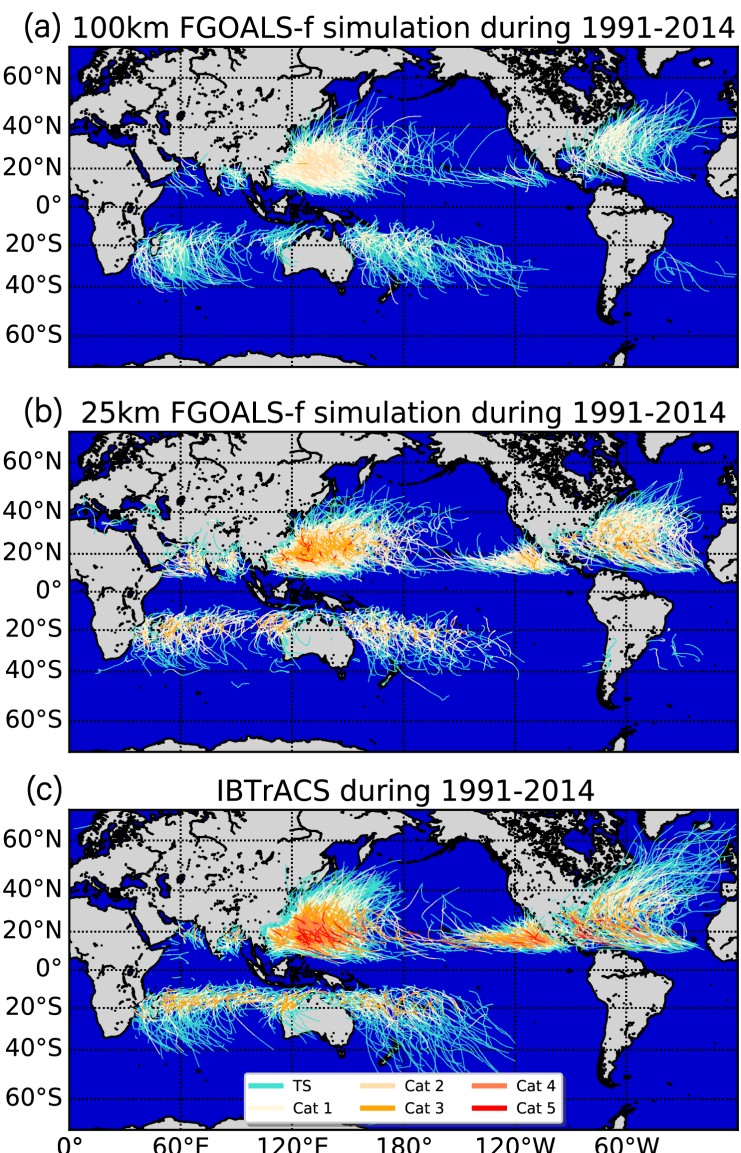

**Figure 1.** Global tropical cyclone tracks (lines) and intensities (colors) during the time period 1991–2014 between (a) FGOALS-f3-L, (b) FGOALS-f3-H and (c) IBTrACS. The tropical cyclones simulated by FGOALS-f3-L/H are picked out using an objective feature-tracking approach and only those tropical cyclones with a lifetime >3 days are shown. The definition of intensity threshold is consistent with the Saffir–Simpson scale: TS, tropical storm; Cat 1–5, category 1–5.


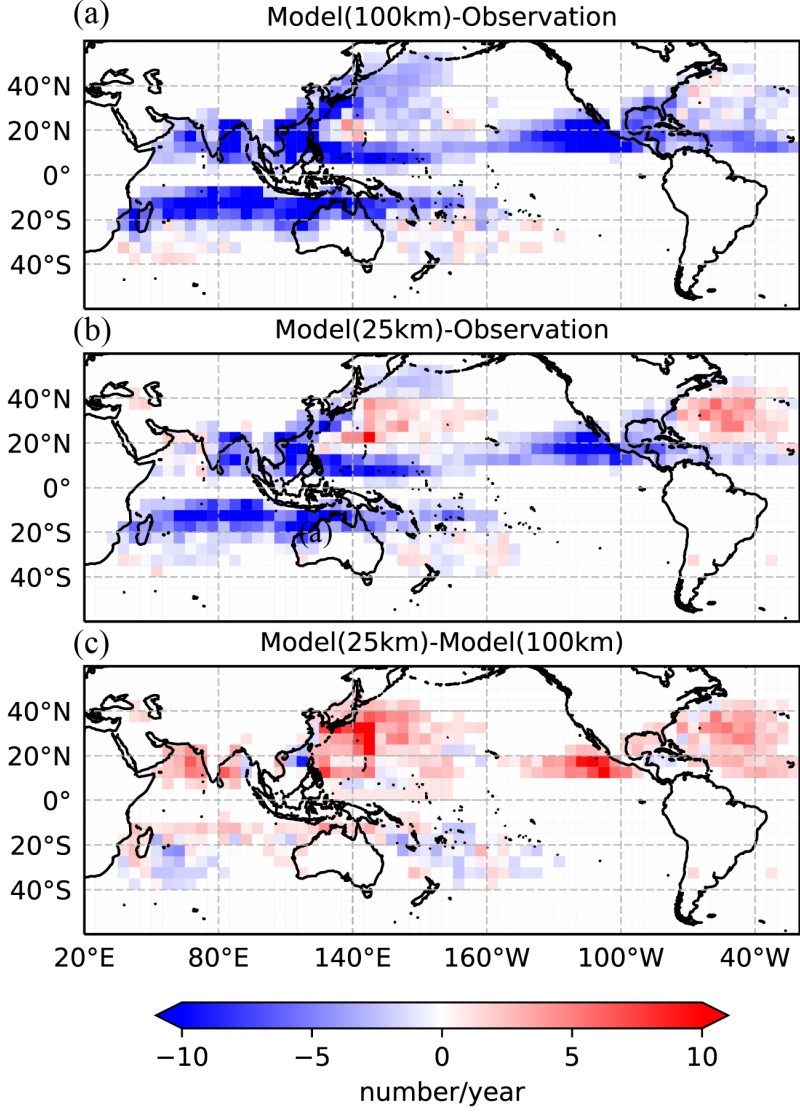

**Figure 2.** Global track density anomalies of tropical cyclones. The track density is analyzed in (5°×5°) equidistance grid boxes at 6-h intervals and the unit of the color map is the number of cyclones per year from 1991 to 2014. (a) Track density anomaly between FGOALS-f3-L and IBTrACS; (b) track density anomaly between FGOALS-f3-H and IBTrACS; and (c) track density anomaly between FGOALS-f3-H and FGOALS-f3-L.




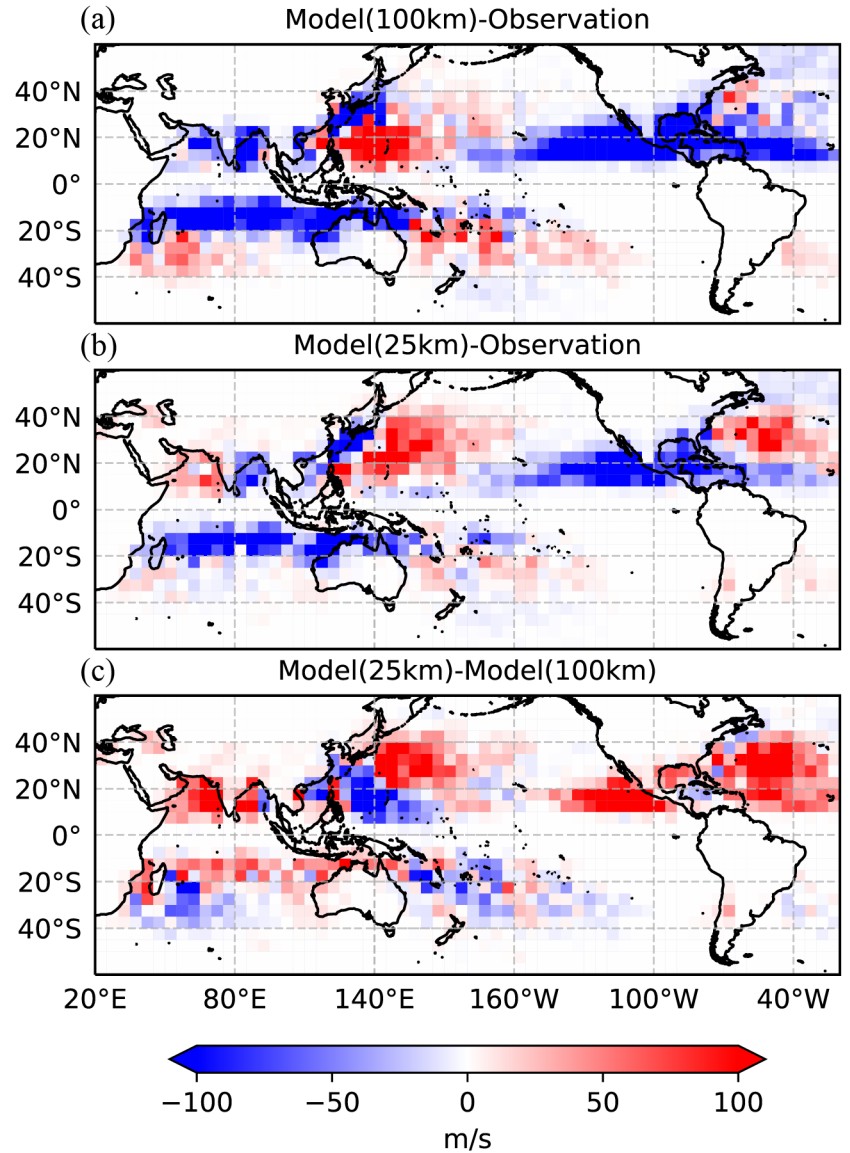

**Figure 3.** Global maximum sustained wind anomalies of the tropical cyclone (units: m s$^{-1}$). The track density is analyzed in (5°×5°) equidistance grid boxes at 6-h intervals. (a) Maximum sustained wind anomalies between FGOALS-f3-L and IBTrACS; (b) maximum sustained wind anomalies between FGOALS-f3-H and IBTrACS; and (c) maximum sustained wind anomalies between FGOALS-f3-H and FGOALS-f3-L.



**Figure 4.** Pressure–wind pairs for each 6-hourly tropical cyclone measurement for FGOALS-f3-L and FGOALS-f3-H (blue dots) and IBTrACS (red dots) in (a, b) the western Pacific Ocean and (c, d) the northern Atlantic Ocean. A linear regression (blue line for FGOALS-f3-L/H; red line for IBTrACS) is fitted to each distribution of pressure–wind pairs.


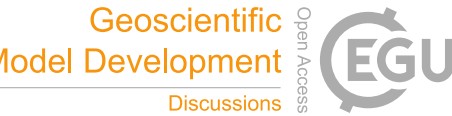

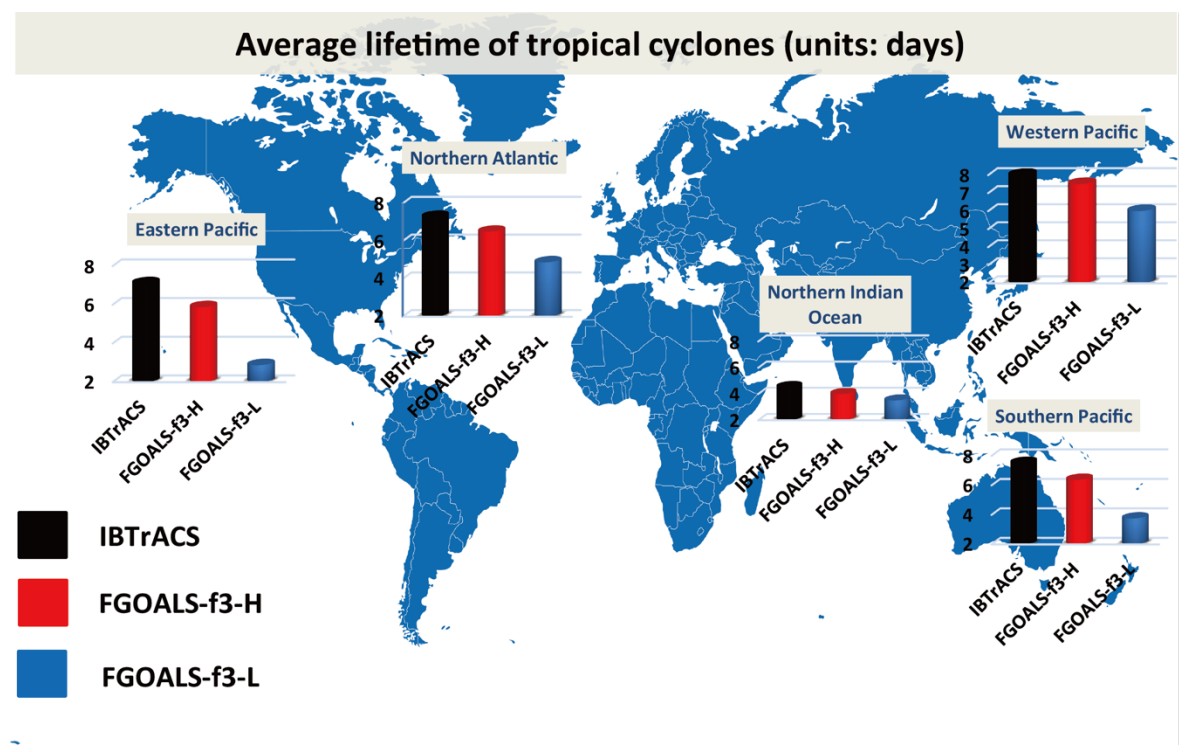

**Figure 5.** Climatological lifetime of tropical cyclones in the western Pacific, southern Pacific, northern Indian, northern Atlantic and eastern Pacific oceans (units: days) during the time period 1991–2014.





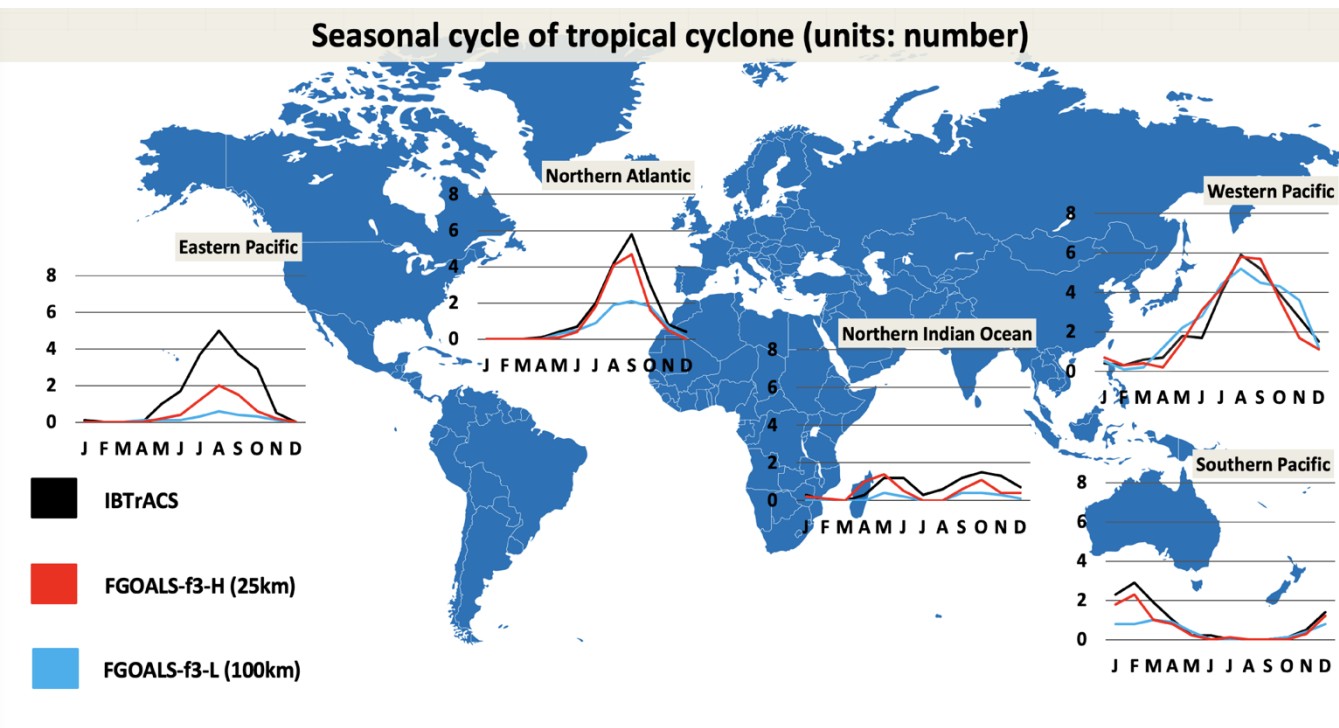

**Figure 6.** Seasonal cycle of tropical cyclones in the western Pacific, southern Pacific, northern Indian, northern Atlantic and eastern Pacific oceans (units: number of cyclones) during the time period 1991–2014.





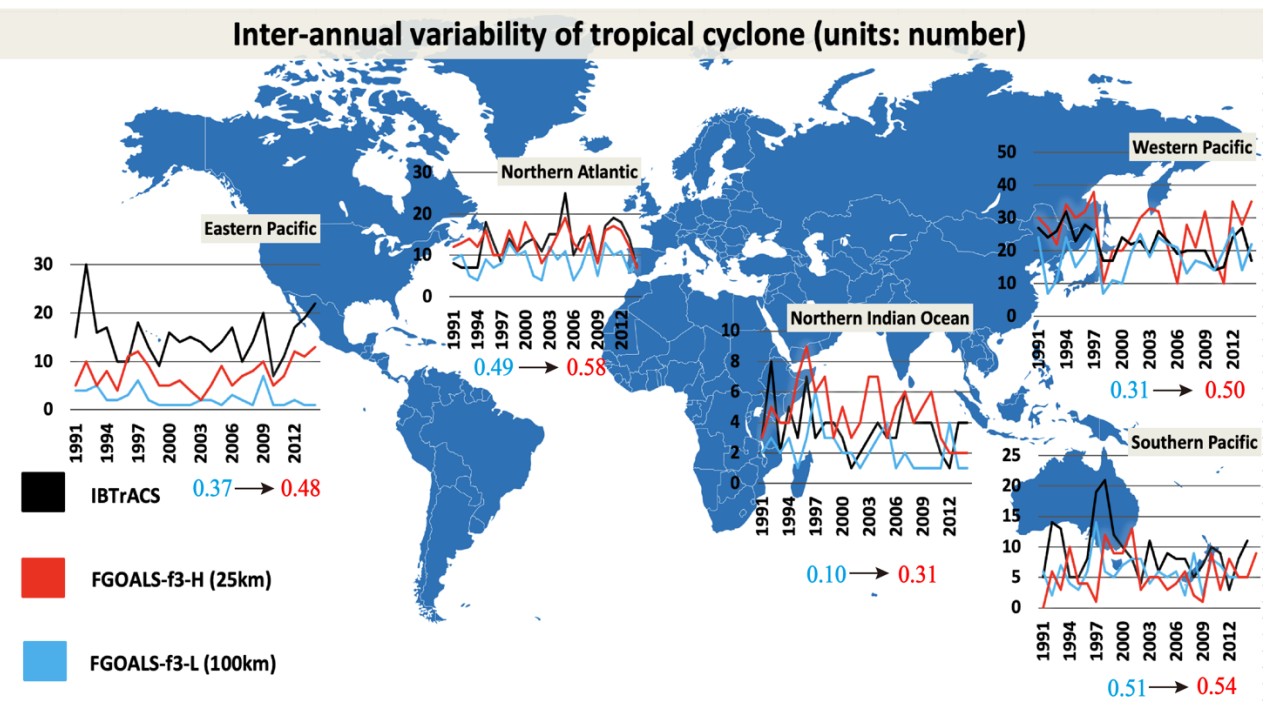

**Figure 7.** Interannual variability of tropical cyclones in the western Pacific, southern Pacific, northern Indian, northern Atlantic and eastern Pacific oceans (units: number of cyclones) during the time period 1991–2014. The numbers under the line charts are the correlation coefficients between IBTrACS and FGOALS-f3.

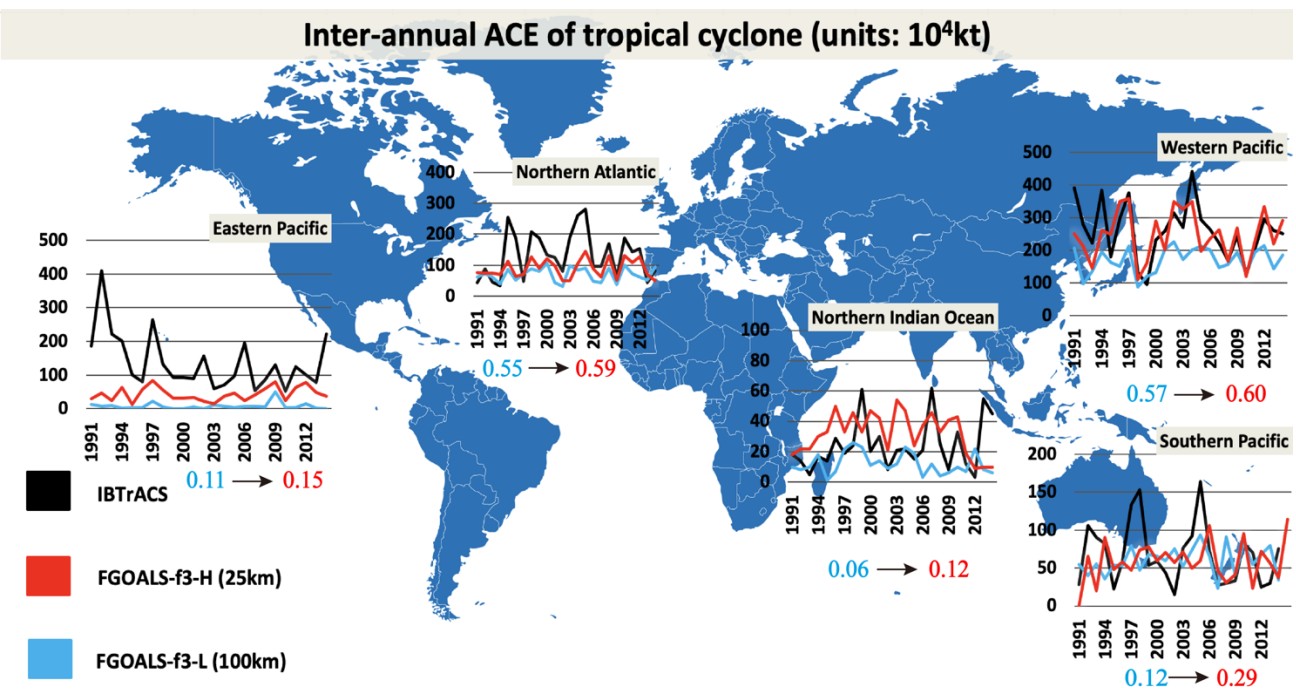

**Figure 8.** ACE of tropical cyclones (units: $10^4$ kt) in the western Pacific, southern Pacific, northern Indian, northern Atlantic and eastern Pacific oceans (units: number of cyclones) during the time period 1991–2014.



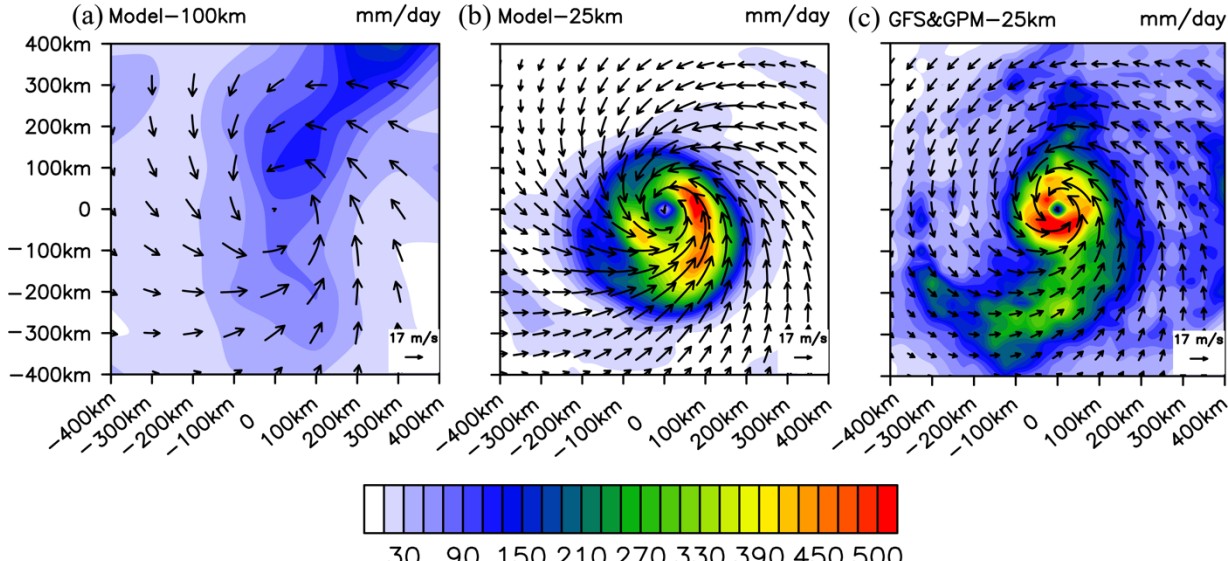

**Figure 9.** Surface 10 m winds (units: m s⁻¹) and daily precipitation rate (mm day⁻¹) for the horizontal composite of the 30 most intense tropical cyclones in (a) FGOALS-f3-L and (b) FGOALS-f3-H. (c) Results of the surface 10 m wind in GFS and the daily precipitation rate in GPM. Radius is 4°.


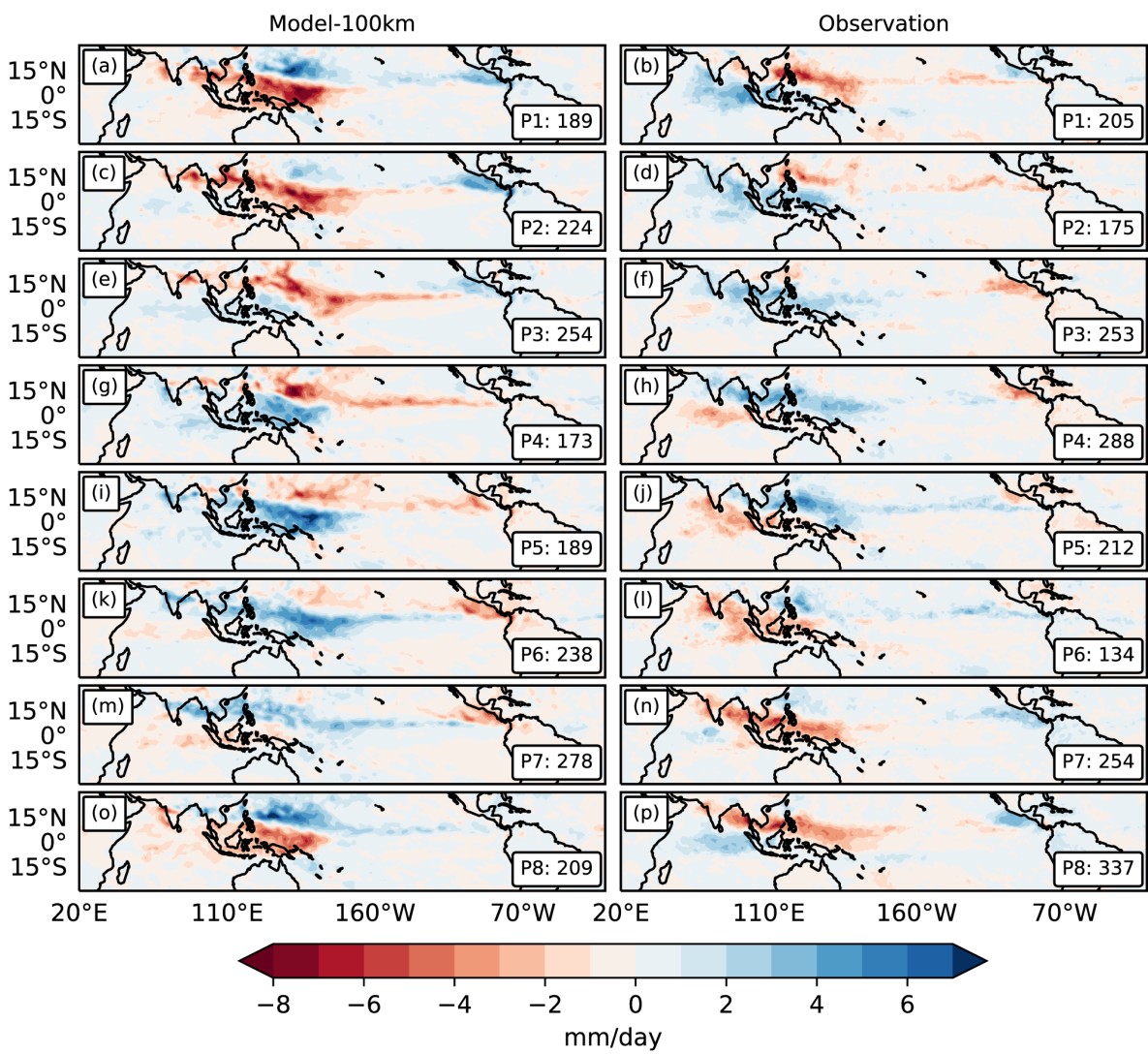

**Figure 10.** Composite May–October 20–100-day precipitation as a function of the phase of the MJO. The method of dividing
the MJO phase follows that of Waliser et al. (2009). The number of days used to generate the composite for each phase is
shown in the lower right-hand corner of each panel. The left-hand side of the panels shows the simulated results derived from
FGOALS-f3-L (a, c, e, g, i, k, m and o) and the right-hand side of the panels shows the simulated results derived from the
Tropical Rainfall Measuring Mission (b, d, f, h, j, l, n and p).



**Figure 11.** Composite May–October 20–100-day precipitation as a function of the phase of the MJO. The method of dividing the MJO phase follows that of Waliser et al. (2009). The number of days used to generate the composite for each phase is shown in the lower right-hand corner of each panel. The left-hand side of the panels shows the simulated results derived from FGOALS-f3-H (a, c, e, g, i, k, m and o) and the right-hand side of the panels shows the simulated results derived from the Tropical Rainfall Measuring Mission (b, d, f, h, j, l, n and p).

**Figure 12.** Daily tropical cyclone track density anomalies (units: $\times 10^1$ number day$^{-1}$) for (top to bottom) MJO phases 1–8. Results are shown for FGOALS-f3-L (left-hand panels) and IBTrACS (right-hand panel) for the time period 1991–2014. The track density is analyzed in (5°×5°) equidistance grid boxes.



**Figure 13.** Daily tropical cyclone track density anomalies (units: $\times 10^1$ number day$^{-1}$) for (top to bottom) MJO phases 1–8. Results are shown for FGOALS-f3-H (left-hand panels) and IBTrACS (right-hand panels) for the time period 1991–2014. The track density is analyzed in (5°×5°) equidistance grid boxes.




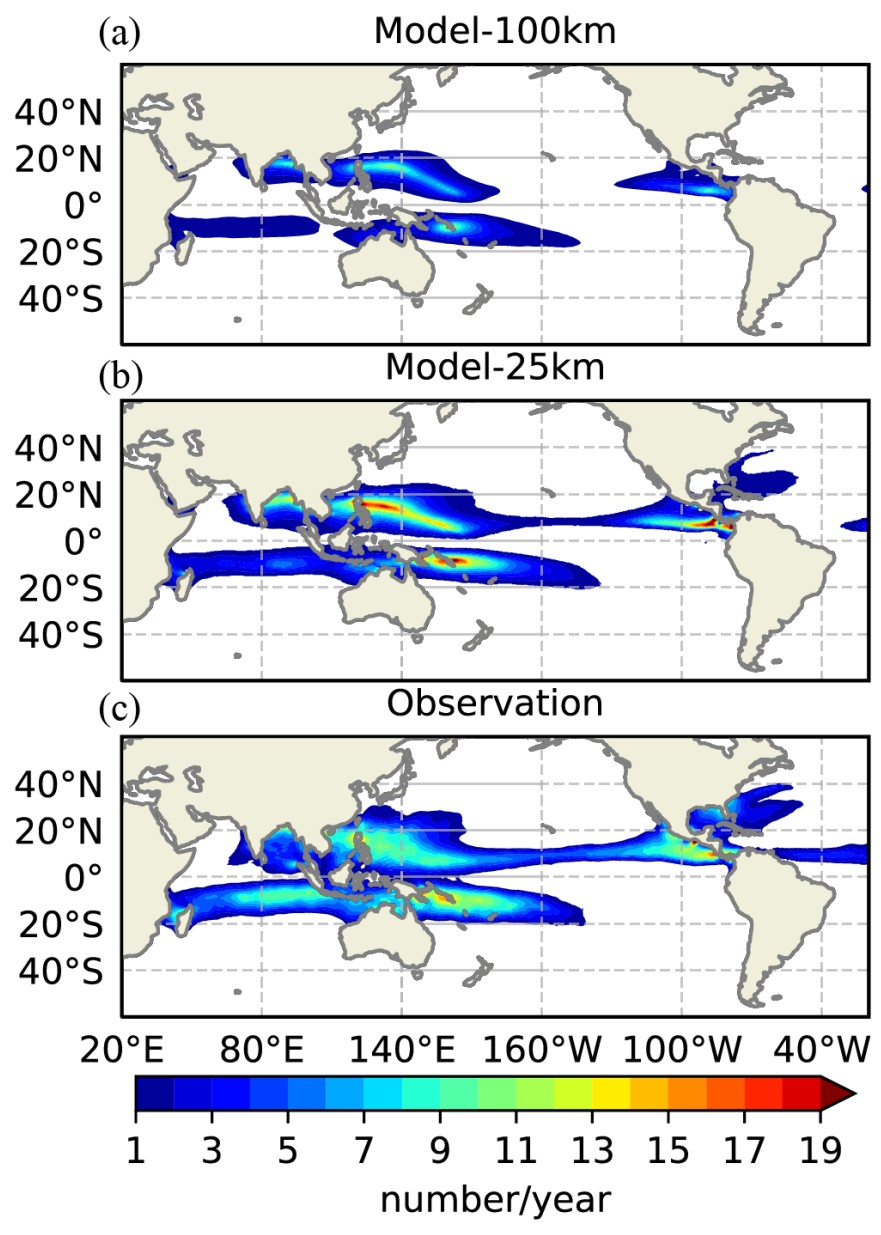

**Figure 14.** Annual tropical cyclone GPI during the time period 1991–2014 (units: number year$^{-1}$), based on the (a) FGOALS-f3-L, (b) FGOALS-f3-H and (c) ERA-Interim datasets.






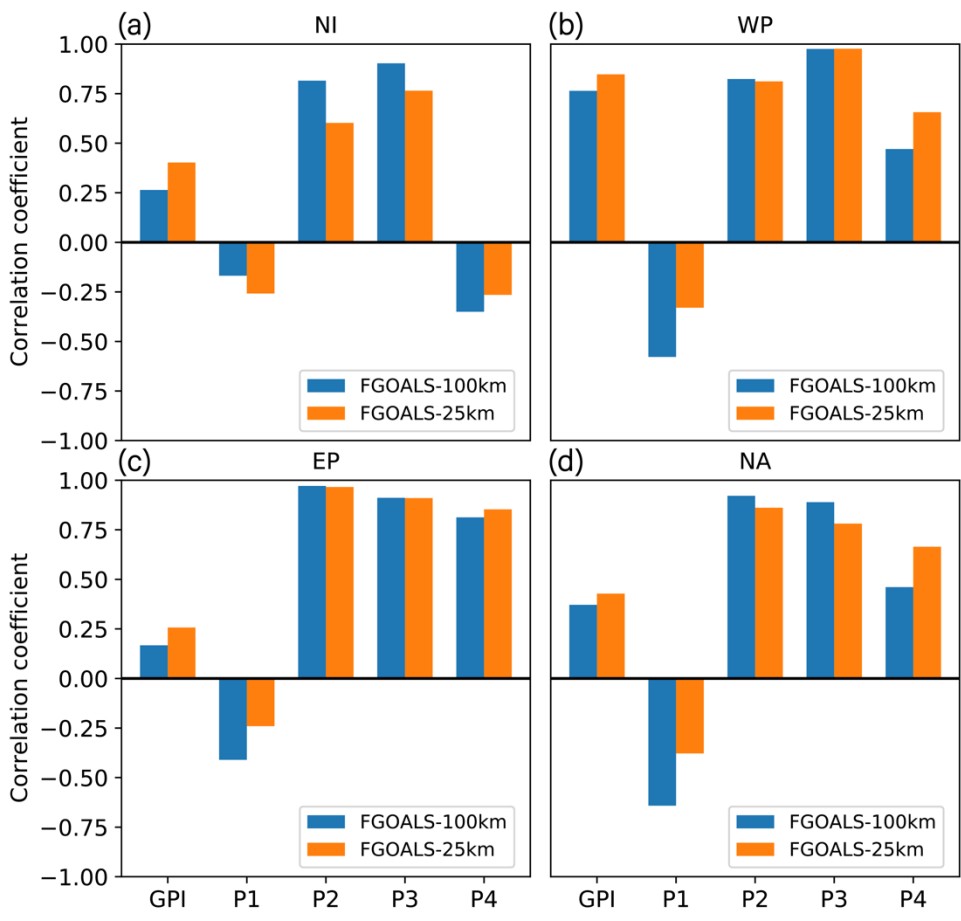

**Figure 15.** Pattern correlation of annual tropical cyclone GPI between the ERA-Interim dataset and the simulation of FGOALS-f3 at low (blue bars) and high (orange bars) horizontal resolutions. P1–P4 represent parts 1–4 of equation 2. P1 represents the equation $|10^5 vort850|^{3/2}$ , P2 represents the equation $\frac{RH}{50}$ , P3 represents the equation $\frac{V_m}{70}$ and P4 represents the equation $(1 + 0.1V_{\text{shear}})^{-2}$.

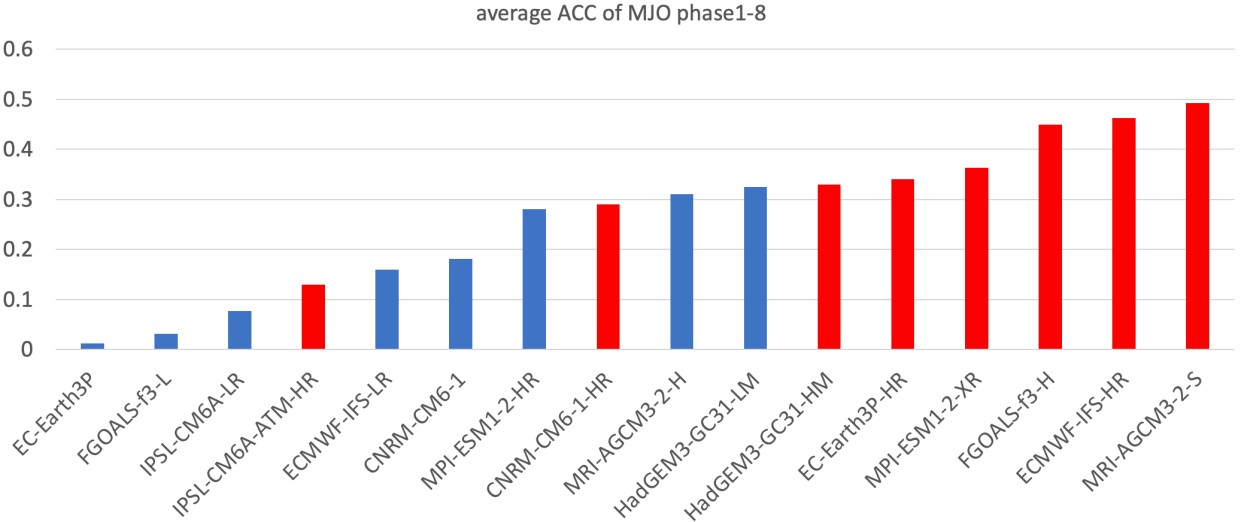

**Figure 16.** Average ACC of the MJO from phases 1 to 8, which composite the May–October 20–100-day precipitation as a function of the MJO. The method of dividing the MJO phase follows that of Waliser et al. (2009). The horizontal axis is the name of the GCMs participating in the HiResMIP for CMIP6 (Haarsma et al. 2016) and the vertical axis is the average ACC. The red cylinders represent the high-resolution GCMs and the blue cylinders represent the low-resolution GCMs.





**Figure 17.** Anomalies of the composite GPI by MJO phases 4–7 between the multi-model mean of the (a) low-resolution GCMs and ERA-Interim and (b) high-resolution GCMs and ERA-Interim. (c) Taylor diagram for the Southern China Sea calculated in the blue box in parts (a) and (b). (d) Taylor diagram in the western north Pacific calculated in the black box in parts (c) and (d).