# Peer review of "Effect of Horizontal Resolution on the Simulation of Tropical Cyclones in the Chinese Academy of Sciences FGOALS-f3 Climate System Model"

_Geoscientific Model Development, 2021_

## Author Comment (AC1)

**Reply to Reviewer #2**

Thank you very much for your interest in FGOALS-f3 and its simulation performance for tropical cyclones activities. Your valuable comments and suggestions have help us to improve the quality of manuscript, and we have learned a lot from your suggestions. The following is our point-by-point reply to your comments.

**Comments:**

This paper serves as documentation of the tropical cyclone activity simulated by the FGOALS-f3 models submitted to the HighResMIP subproject. While there is little unexpected in the comparison of low to high resolution models, it is important to have such individual model results in the literature. I recommend that it be sent to the authors for some fairly minor revisions that I describe in detail below.

1. **Section 3.1:** Figure 2. It is difficult to synthesize by eye the biases in figure 2. I would like to see either a bar chart figure or a table with observed and simulated TC counts both globally and by ocean basin.

   Thank you for your valuable suggestion. We have added a table (Table 4) showing the observed and simulated tropical cyclone counts, both globally and by ocean basin.

Table 4. Observed and simulated average tropical cyclone number, both globally and by ocean basin, in the northern Indian (NI), western Pacific (WP), eastern Pacific (EP), northern Atlantic (NA), southern Indian Ocean (SI), southern Pacific (SP) and southern Atlantic (SA) oceans.

| Data source | Global | NI | WP | EP | NA | SI | SP | SA |
|---|---|---|---|---|---|---|---|---|
| IBTrACS | 82.67 | 4.05 | 26.24 | 15.00 | 13.85 | 14.25 | 9.14 | 0.14 |
| FGOALS-f3-L | 53.14 | 1.98 | 25.04 | 3.96 | 7.54 | 7.34 | 6.83 | 0.45 |
| FGOALS-f3-H | 67.72 | 3.25 | 27.46 | 10.00 | 11.83 | 8.63 | 6.09 | 0.46 |

2. **Figure 4.** Please note that the bias in the min pressure/max wind speed is worse in the North Atlantic than in the western Pacific in the high-resolution model. Why is this?

Thank you for your question. As shown in Figure 4, the bias in tropical cyclone intensities in the NA are greater than those in the WP. We did not tune the model for specific regions. However, to comply with the HighResMIP rule "The experimental set-up and design of the standard resolution experiments will be exactly the same as for the high-resolution runs", we tried to keep the model setting consistent when the horizontal resolution was increased from 100 to 25 km. It is possible that the strong tropical cyclone event in the NA is still not well resolved at 25 km resolution in FGOALS-f3. There was still a negative bias in the tropical cyclone count in the NA when the horizontal resolution increased from 100 to 25 km. Another possible reason is related to the Resolving Convective Precipitation (RCP) scheme (Bao and Li, 2020) used in FGOALS-f3. The RCP scheme calculates convective and stratiform precipitation at the grid scale, which is clearly different from the traditional convective parameterization. Current studies indicate that the sub-grid parameterization in convective schemes is sensitive to the simulated intensities of tropical cyclones even when the horizontal resolution of GCMs is increased to 25 km (Murakami et al., 2012; Lim et al., 2015). We think that FGOALS-f3 with the RCP scheme does not give the best performance at 25 km and it is worth increasing the horizontal resolution (e.g., 1/8°) to verify this assumption.

References:

Bao, Q. and Li, J.: Progress in climate modeling of precipitation over the Tibetan Plateau, Natl. Sci. Rev., 7, 486-487, https://doi.org/10.1093/nsr/nwaa006, 2020.

Murakami H, Wang Y, Yoshimura H, et al. Future changes in tropical cyclone activity projected by the new high-resolution MRI-AGCM. Journal of Climate, 2012, 25(9): 3237-3260.

Lim Y K, Schubert S D, Reale O, et al. Sensitivity of tropical cyclones to parameterized convection in the NASA GEOS-5 model[J]. Journal of Climate, 2015, 28(2): 551-573.

3. **Lines213-215**: "Neither the single peak in the number of tropical cyclones in the northern Atlantic (peak month September), eastern Pacific (peak month August) and southern Pacific (peak month February) oceans nor the double peak in the northern Indian Ocean (peak months May and November) could be reproduced in FGOALS-f3-L." I don't think this is actually correct, although the low-resolution model does not produce the magnitude of these peaks, it does appear to replicate the timing of the seasonal cycle. This would be more apparent by normalizing figure 6 by the number of storms per basin. Admittedly, the Southern Pacific does appear to be delayed.

Thank for your valuable suggestion. We agree with your view about the seasonal cycle of the tropical cyclone number. Although FGOALS-f3-H generates more tropical cyclone counts in each basin, FGOALS-f3-L replicates the timing of the seasonal cycle. The seasonal cycle with normalized tropical cyclone counts is a good method by which to compare the simulation of the seasonal cycle of tropical cyclones between FGOALS-f3-L and FGOALS-f3-H. So, as suggested, we have added a figure in the supplementary material to show the normalized seasonal cycle of tropical cyclones in each basin. We have added the sentence as "Although FGOALS-f3-H can produce more tropical cyclone counts in the peak month in each basin, both FGOALS-f3-L and FGOALS-f3-H appeared to replicate the timing of the seasonal cycle when we normalized the results of the tropical cyclone seasonal cycle (Figure S1)" at lines 233–235.

[Figure]

**Figure S1.** Seasonal cycle of tropical cyclones with zero-mean normalization in the western Pacific, southern Pacific, northern Indian, northern Atlantic and eastern Pacific oceans (units: number of cyclones) during the time period 1991–2014.

4. **Lines 226-229:** It is a bit of a stretch to claim that the interannual correlation of ACE is improved with resolution in WP and NA as the differences are very small in figure 9. In fact, given that the correlation in interannual counts changes a fair amount in figure 8, one might expect that the ACE correlation should change even more, given the dependence on the square of peak wind speed and the differences in that field between resolutions. A more interesting quantity might be simply the average ACE per basin.

The average accumulated cyclone energy (ACE) is an interesting quantity with which to compare the intensity of tropical cyclones in each basin. As suggested, we have added a table (Table 5) to show the average ACE between the observations and simulations.

Table 5. Observed and simulated averaged ACE (units: 104 kt) in the northern Indian (NI), western Pacific, eastern Pacific, northern Atlantic (NA), southern Indian (SI), southern Pacific (SP) and southern Atlantic (SA) oceans.

| Data source | NI | WP | EP | NA | SP |
|---|---|---|---|---|---|
| IBTrACS | 24.21 | 258.75 | 137.42 | 133.13 | 67.58 |

| | | | | | |
|---|---|---|---|---|---|
| FGOALS-f3-L | 12.13 | 170.47 | 7.83 | 69.38 | 60.30 |
| FGOALS-f3-H | 32.08 | 247.66 | 43.66 | 89.10 | 61.21 |

5. **Section 3.3:** Grammar. Instead of "The extreme position of precipitation", you mean "The position of extreme precipitation". Figure 9 is quite interesting. It may be clearer to express the bias in terms of an angle and radial distance. It does appear that the radial distance is quite good. Any thoughts on the error in angle? Also, the diameter of the eye would appear to be only one or two grid cells. It should be mentioned that although an eyewall is present, it is not resolved at this resolution

Thank you for your correction. "The position of extreme precipitation" is the correct meaning and we have changed this sentence from "The extreme position of precipitation" to "The position of extreme precipitation" at line 255. Chen et al. (2006) found that the vertical wind shear and storm motion are the two most important factors contributing to asymmetries in rainfall in tropical cyclones. We therefore think the error in the angle is due to the biases in the wind shear and storm motion when the intensity of the tropical cyclone reaches a maximum in FGOALS-f3. The non-hydrostatic dynamical core used in FGOALS-f3 and the limited air–sea coupling processes (Kim et al., 2018) (AMIP) also contribute to this error. We have therefore modified the statement at lines 262–266 to "Chen et al. (2006) found that the vertical wind shear and storm motion are the two most important factors contributing to rainfall asymmetries in tropical cyclones. The biases in the vertical wind shear and storm motion in FGOALS-f3 may affect the angle of the horizontal structure of tropical cyclones. The non-hydrostatic dynamical core used in FGOALS-f3 and the limited air–sea coupling processes (Emanuel et al., 2013; Kim et al., 2018) (AMIP) also contribute to the error".

References:

Chen, Shuyi S., John A. Knaff, and Frank D. Marks Jr. "Effects of vertical wind shear and storm motion on tropical cyclone rainfall asymmetries deduced from TRMM." Monthly Weather Review, 2006, 134.11: 3190-3208.

Kim D, Ho C H, Park D S R, et al. The relationship between tropical cyclone rainfall area and environmental conditions over the subtropical oceans. Journal of Climate, 2018, 31(12): 4605-4616

Emanuel, K., and Sobel, A. Response of tropical sea surface temperature, precipitation, and tropical cyclone-related variables to changes in global and local forcing, Journal of Advances in Modeling Earth Systems, 2013, 5(2), 447-458.

6. **Section 4:** In my view, the biggest source of difference between TC activity in the two models comes from the storm tracker. Despite the threshold adjustment table 3 (which is quite small), the trackers such as used here (or in TempestExtremes) are generally going to miss the weak storms in the low-resolution models. Trackers such as TRACK show much higher storm counts in low resolution models (see Roberts et al.). So, while the improvements in MJO and GPI are interesting, it is hard to claim that they are responsible for the higher TC counts when there is such a strong dependence on the choice of storm tracker. You may consider shortening these sections.

Thank you for your valuable suggestion. Although we examined the sensitivity of thresholds in our tracker, which is not very sensitive to the threshold of wind speed and vorticity, we did not take into account the biases of different trackers (e.g., TempestExtremes, TRACK and TSTORM). It is a good idea to reduce the uncertainty in the recognition of tropical cyclones in GCMs. The work of Roberts et al. (2020a; 2020b) is outstanding in revealing the errors caused by different methods in the simulation and projection of tropical cyclones. As suggested, we have added a discussion about the different trackers at lines 336–339: "However, the difference between the tracking algorithms—such as TRACK (Hodges et al.,

2017), TSTORM (Zhao et al., 2009) and TempestExtremes (Ullrich et al., 2017, 2021)—are also an important factor in the uncertainties in tropical cyclone simulations. Cross-validation of the performance of tropical cyclone simulations with multiple tracking algorithms is necessary in future research (Roberts et al., 2020)". We agree with your suggestion that the GPI and MJO are not unique, dominant factors. We have therefore rewritten these sections of the paper.

References:

Hodges, Kevin, Alison Cobb, and Pier Luigi Vidale. "How well are tropical cyclones represented in reanalysis datasets?." Journal of Climate 30.14 (2017): 5243-5264.

Roberts, Malcolm John, et al. "Projected future changes in tropical cyclones using the CMIP6 HighResMIP multimodel ensemble." Geophysical research letters 47.14 (2020a): e2020GL088662.

Roberts, Malcolm John, et al. "Impact of model resolution on tropical cyclone simulation using the HighResMIP–PRIMAVERA multimodel ensemble." Journal of Climate 33.7 (2020b): 2557-2583.

---

## Author Comment (AC2)

**Reply to Reviewer #1**

Thank you very much for your careful review, detailed comments and constructive suggestions, which have helped greatly to improve the quality of our paper. The following is our point-by-point reply to your comments.

**Comments:**

In this study, the authors present the simulation of TCs in the FGOALS-f3 climate model. They use simulations at two different spatial resolutions to understand the impact of model resolution on TC simulation. I find the presentation not very good and highly disorganized. Also, the explanations provided are very rushed and hand-wavy. There's an over-emphasis on the role of MJO in TCs without first considering the basic large-scale environmental parameters governing TC formation and development first.

1. **Line 25:** Not sure what you mean by 'seasonal cycle of number of tropical cyclones increased by 50%'.

   Thank you for your careful correction. We have modified the statement from "seasonal cycle of number of tropical cyclones increased by 50%" to "Although the number of tropical cyclones increased by about 50% at the higher resolution and better matched the observed values in the peak month, both FGOALS-f3-L and FGOALS-f3-H appear to replicate the timing of the seasonal cycle of tropical cyclones." at lines 27–28.

2. **Line 55:** Replace 'following half-century' with 'last few decades'

   Thank you for your suggestion. We have replaced the statement from "following half-century" to "last few decades" at line 55.

3. **Line 60:** Cite past studies that have used high-resolution coupled GCMs to simulate TCs (Kim et al., 2014; Small et al., 2014; Li and Sriver 2018; Scoccimarro et al., 2017; Balaguru et al., 2020).

We are pleased to cite (at line 60-61) the following studies that have used high-resolution coupled GCMs to simulate tropical cyclones.

References:

Kim, Hyeong-Seog, et al. "Tropical cyclone simulation and response to $CO_2$ doubling in the GFDL CM2. 5 high-resolution coupled climate model." Journal of Climate 27.21 (2014): 8034-8054.

Small, R. Justin, et al. "A new synoptic scale resolving global climate simulation using the Community Earth System Model." Journal of Advances in Modeling Earth Systems 6.4 (2014): 1065-1094.

Li, Hui, and Ryan L. Sriver. "Tropical cyclone activity in the high-resolution community earth system model and the impact of ocean coupling." Journal of Advances in Modeling Earth Systems 10.1 (2018): 165-186.

Scoccimarro, E., et al. "Tropical cyclone interaction with the ocean: The role of high-frequency (subdaily) coupled processes." Journal of Climate 30.1 (2017): 145-162.

Balaguru, Karthik, et al. "Pronounced impact of salinity on rapidly intensifying tropical cyclones." Bulletin of the American Meteorological Society 101.9 (2020): E1497-E1511.

4. **Line 80:** Replace 'controversial' with 'ambiguous'

   Thank you for your suggestion. We have replaced 'controversial' with 'ambiguous' at line 87.

5. **Lines 67-87:** This part could be better written and organized.

   Thank you for your suggestion. We have rewritten lines 68–92 as: "The increase in the horizontal resolution of GCMs has led to significant changes in the simulation

of the variability of tropical cyclones. The changes can be broadly attributed to two reasons: (1) changes in the large-scale factors; and (2) the development of physical process parameterization and air–sea coupling related to the simulation of tropical cyclones. High-resolution GCMs need to not only give a better description of the structure of tropical cyclones, but should also simulate well the relationship between tropical cyclones and large-scale variabilities—for example, the El Niño–Southern Oscillation (ENSO), the Madden–Julian oscillation, wind shear and vorticity, and humidity—which is crucial in reducing the uncertainties in the simulation and prediction of tropical cyclones (Manganello et al., 2012, 2016; Zhang et al., 2016; Delworth et al., 2020). Previous studies have shown that there are significant changes in the ENSO as the horizontal resolution of GCMs increases (Philander et al., 1992; Kuntson et al., 1997; Schneider et al., 2003; Masson et al., 2012; Larson et al., 2013; Meehl et al., 2020) and the simulation results are mostly positive. However, these improvements in predicting the ENSO with an increase in horizontal resolution did not lead to improvements in the relationship between the ENSO and tropical cyclones (Matsuura et al., 1999; Bell et al., 2014; Krishnamurthy et al., 2016). There is also a relationship between the Madden–Julian oscillation and tropical cyclones (Liebmann et al., 1994; Hall et al., 2001; Camargo et al., 2008, 2009; Zhang et al., 2013; Klotzbach et al., 2014).

As the horizontal resolution in the models increases, some key parameters in the physical parameterizations are tuned to give a better performance (Bacmeister et al., 2013; Roberts et al., 2020)—for example, Lim et al. (2015) found that an increase in the threshold of minimum entrainment led to increasing tropical cyclone activity and Murakami et al. (2012) found that the constrained convective heating in the convective scheme induced intense grid-scale upward motion and promoted large-scale condensation, which favored the development of a more intense tropical cyclone. These artificial tuning methods might introduce more uncertainties in terms of the effects of resolution, giving rise to conclusions that are ambiguous to the tropical cyclone research community. In addition, considering the air–sea

coupling process will also affect the simulation performance of tropical cyclone activities, especially with respect to the intensity. Scoccimarro et al. (2017) found that an increased horizontal resolution of the model components was not sufficient to simulate intense and fast-moving tropical cyclone events and that air–sea coupling with a higher coupling frequency helps to improve the performance of simulations of tropical cyclone intensity".

6. **Lines 110-120:** While the atmospheric component is described in detail, there is only a one-line statement about the other components, especially the ocean model. Since this is a coupled simulation, and TC development is a highly coupled phenomenon, the authors must provide details of the ocean model as well.

Thank you for your suggestion. The ocean model is an important component of GCMs. As suggested, we have added an introduction to the ocean model used in FGOALS-f3 at lines 124–126: "The oceanic component is the LASG/IAP Climate System Ocean Model Version 3 (LICOM3) (Liu et al., 2012). Orthogonal curvilinear coordinates and a tripolar grid are used in LICOM3 and the horizontal resolution can vary flexibly between 1o and 1/20o. A new advection scheme has also been updated in LICOM3 (Yu et al., 2018)".

Reference:

Yu, Y., Tang, S., Liu, H., Lin, P., and Li, X.: Development and evaluation of the dynamic framework of an ocean general circulation model with arbitrary orthogonal curvilinear coordinate, Chinese Journal of Atmospheric Sciences, 42, 877–889, https://doi.org/10.3878/j.issn.1006-9895.1805.17284, 2018.

7. **Lines 180-185:** Is this due to positive SST biases in higher latitudes? It could also be the effect of steering flow being too strong in the North Atlantic and Northwest Pacific, which prevent TCs from making landfall.

Thank you for your suggestion. Because we only planned to include Tier 1 and Tier 2 of HighResMIP, which use a high-resolution SST, to force the atmospheric model, we carried out an AMIP-like run. As a result, the SST biases may not be considered in our results. The biases in the large-scale factors (e.g., the strong steering flow) related to the tropical cyclones in GCMs may contribute to the simulated biases in tropical cyclone activity in the WP and NA. So, as suggested, we have modified the sentence at lines 196–199 to: "This phenomenon also exists in the high-resolution GCMs that participated in the European Union Horizon 2020 project PRIMAVERA (Roberts et al., 2020). The biases in the large-scale factors (e.g., strong steering flow) related to the tropical cyclones in GCMs may lead to the simulated biases of tropical cyclone activities in the western Pacific and northern Atlantic oceans".

References:

Haarsma, R., Acosta, M., Bakhshi, R., et al.: HighResMIP versions of EC-Earth: EC-Earth3P and EC-Earth3P-HR–description, model computational performance and basic validation, Geosci. Model Dev., 13, 3507-3527, https://doi.org/10.5194/gmd-13-3507-2020, 2020.

Roberts, M. J., Camp, J., Seddon, J., Vidale, P. L., Hodges, K., Vanniere, B., Mecking, J., Haarsma, R., Bellucci, A., and Scoccimarro, E.: Impact of Model Resolution on Tropical Cyclone Simulation Using the HighResMIP–PRIMAVERA Multimodel Ensemble, J. Climate, 33, 2557-2583, https://doi.org/10.1175/JCLI-D-19-0639.1, 2020.

Roberts, C. D., Senan, R., Molteni, F., Boussetta, S., Mayer, M., and Keeley, S.: Climate model configurations of the ECMWF integrated forecast system (ECMWF-IFS cycle 43r1) for HighResMIP, Geosci. Model Dev., 11, 3681–3712, https://doi.org/10.5194/gmd-11-3681-2018, 2018.

8. **Figure 4:** Why focus on only North Atlantic and Northwest Pacific in a GCM? To me, the improvement obtained going from a 100 km model to a 25 km model is

obvious and not very interesting. The authors can focus more on their results based on the high-resolution model and show global results instead of focusing on a couple of basins.

Thank you for your valuable suggestion. We agree with your view that it is not sufficient to show only the tropical cyclone intensities in the WP and NA. It is more interesting to show the intensities of tropical cyclones in each oceanic basin around the world. As suggested, we have modified Figure 4 to show the pressure–wind pairs in each ocean basin.

[Figure]

**Figure 4.** Pressure–wind pairs for each 6-hourly tropical cyclone measurement for FGOALS-f3-L (blue dots) and FGOALS-f3-H (red dots) and IBTrACS (black dots) in (a) northern Indian Ocean, (b) western Pacific, (c) eastern Pacific, (d) northern Atlantic and (e) southern Pacific. A linear regression (blue/red line for FGOALS-f3-L/H; black line for IBTrACS) is fitted to each distribution of pressure–wind pairs.

9. **Line 200:** The biggest increase in TC duration appears to be in the eastern Pacific. Why is this the case? Also, why is there an increase in TC lifetimes in general? Is it because of an increase in intensity? Or is it because of biases in steering flow?

Thank you for your question. On one hand, the increased horizontal resolution in our convective precipitation scheme contributes to improvements in simulating the lifetime of tropical cyclones in general and the tropical cyclone counts and intensities are significantly increased. On the other hand, the biases in the large-scale factors (Figure 11) in FGOALS-f3-H are reduced (e.g., the wind shear), which contributes to the generation and development of tropical cyclones.

10. **Lines 210-220:** Why doesn't the seasonal cycle improve much in the eastern Pacific unlike every other basin?

Thank you for your question. Few tropical cyclone counts have been identified in FGOALS-f3-L. When the horizontal resolution increased from 100 to 25 km, the negatives biases in the tropical cyclones in the EP and NA were improved. Both versions of the model retained the exact model physics and parameters and the only differences were the horizontal resolution and model time steps, which better met the rule of the HighResMIP. So, we think that FGOALS-f3-H at 25 km horizontal resolution is still not capable of capturing all the tropical cyclone activity in the EP and NA without tuning of the physical process parameterization. It is worth continuing to increase the horizontal resolution (C768; ~12.5 km). There are biases in the large-scale factors related to tropical cyclones in the EP and NA, which affected the generation and development of tropical cyclones.

11. **Figures 7 and 8:** If the simulation is free-running and not forced, I'm not sure what the point is in this comparison. In fact, I'd say this is meaningless. The only thing perhaps one can compare is standard-deviation or a measure of interannual variability.

Thank you for your question. According to the requirement of the HighResMIP, the experiment is running with an AMIP-like setting forced by the high-resolution SST.

Reference:

Haarsma, Reindert J., et al. "High resolution model intercomparison project (HighResMIP v1. 0) for CMIP6." Geoscientific Model Development 9.11 (2016): 4185-4208.

12. **Figure 9:** For observations, are you using the most intense TCs?

Thank you for your question. We used the most intense tropical cyclones for the observations to compare the simulated performance of the most intense tropical cyclone in FGOALS-f3.

13. **Section 4.1:** The jump from the previous section to this is rather sudden. I suggest presenting the analysis of the large-scale environment first before getting into MJO, ENSO etc. Note that these phenomena only modulate TC activity.

Thank you for your valuable advice. We agree with your view that the jump from the simulation performance of tropical cyclones to the MJO is rather sudden. The central aim of this paper is to introduce the CMIP6 version of FGOALS-f3 and its simulated performance of global tropical cyclone activity. The MJO is just one of the possible dynamical reasons we used to explain our results. As suggested, we have shortened the introduction of the relationship between the MJO and tropical cyclone activity because there are still some uncertain effects between them. As suggested, we now focus on the relationship between the ENSO and tropical cyclones. We have therefore modified Section 4.1 as:

4 Possible reasons for the simulated performance of tropical cyclones in FGOALS-f3

4.1 Modulation of tropical cyclone activity by the ENSO

There is a lot of evidence to suggest that the ENSO modulates the activity of tropical cyclones. Gray et al. (1984) found that tropical cyclone counts in the Atlantic Ocean are modulated by the ENSO. El Niño (La Niña) events enhanced (suppressed) westerly winds and led to stronger (weaker) vertical wind shear in the Atlantic basin, leading to an increase (decrease) in tropical cyclone counts. Camargo et al. (2004) found that the ACE in the western Pacific is positively correlated with ENSO indices. There are more intense and longer lived tropical cyclones in El Niño years than in La Niña years. Kim et al. (2011) found that the ENSO modulates tropical cyclone activity in the eastern Pacific Ocean. The track densities and genesis of tropical cyclones tend to be enhanced (suppressed) in eastern Pacific warming (cooling) years by strong (weak) westerly wind shear.

Figure 10 shows the average number of tropical cyclones and the ACE from El Niño, neutral and La Niña years. In the western Pacific basin (Figure 10a), there is no clear change in tropical cyclone counts compared with the variation of the ACE between El Niño and La Niña years. FGOALS-f3-H can capture these features in the observations; in particular, the simulation of the ACE is better than in FGOALS-f3-L. In the eastern Pacific basin (Figure 10b), FGOALS-f3 can capture the variation in tropical cyclone activities from El Niño to La Niña years, but the decreasing trend of tropical cyclone counts and the ACE in FGOALS-f3-L/H is weaker than in the observations. In the observations for the northern Atlantic basin (Figure 10c), there are more intense tropical cyclone events in La Niña years. FGOALS-f3 can reproduce the impact of the ENSO on tropical cyclone activity in the northern Atlantic Ocean and the simulated performance of tropical cyclones in FGOALS-f3-H is better than that in FGOALS-f3-L.

[Figure]

**Figure 10.** Bar chart showing the average number of tropical cyclones (left-hand panels) and ACE (right-hand panels) from El Niño (EL), neutral (NE) and La Niña (LA) years in the (a) western Pacific (WP), (b) eastern Pacific (EP) and (c) northern Atlantic (NA) oceans.

**References:**

Kim, Hye-Mi, Peter J. Webster, and Judith A. Curry. "Modulation of North Pacific tropical cyclone activity by three phases of ENSO." Journal of Climate 24.6 (2011): 1839-1849.

Camargo, Suzana J., and Adam H. Sobel. "Western North Pacific tropical cyclone intensity and ENSO." Journal of Climate 18.15 (2005): 2996-3006.

Gray, William M. "Atlantic seasonal hurricane frequency. Part I: El Niño and 30 mb quasi-biennial oscillation influences." Monthly Weather Review 112.9 (1984): 1649-1668.

14. **Figures 10-13:** While figures 10 and 11 do show that the high-resolution model has a better representation of the MJO, its connection to TCs is very hand-wavy and not clear to me. Also, in both figures 12 and 13, MJO seems to have little effect on TCs in the Southern Hemisphere, which is strange. If the authors are really keen on understanding the impact of MJO simulation on TCs, they should perform an analysis something like that shown in this study: https://journals.ametsoc.org/view/journals/clim/27/6/jcli-d-13-00483.1.xml

Thank you for your recommendation of this study, which discusses the impact of the MJO on tropical cyclone activities. We have read it and learned a lot. The main purpose of our paper is to introduce the FGOALS-f3 GCMs participating in

HighResMIP and to show the simulated performance of tropical cyclones in FGOALS-f3 at 100 and 25km horizontal resolution. As suggested, we have rewritten Section 4.1 to discuss the impact of the ENSO on tropical cyclone activity instead of the impact of the MJO.

Reference:

Klotzbach, P. J.: The Madden–Julian oscillation's impacts on worldwide tropical cyclone activity, J. Climate, 27, 2317-2330, https://doi.org/10.1175/JCLI-D-13-00483.1, 2014.

15. **Figure 14:** Expand the domain in the Atlantic all the way to the African coast and add panels for differences with observations. Although the GPI analysis is good, the way it is presented is not helping much. For instance, why is there a tendency in the model for a poleward shift in TCs? It's hard to see anything in the GPI analysis. What about SST biases?

Thank for your valuable advice. We quite agree with your view. We have shortened the relevant introduction of the GPI. As an alternative, we analyzed the large-scale factors associated with tropical cyclone activity (Figure 11). The improvement in the large-scale factors may contribute to the simulated performance of tropical cyclones in FGOALS-f3-H. We have also expanded the domain related to the NA to cover all of the basin.

[Figure]

**Figure 11.** Biases in the large-scale environmental factors related to tropical cyclone activity between FGOALS-f3 and the observations. (a, b) Relative humidity biases at 600 hPa; (c, d) absolute vorticity biases at 850 hPa; (e, f) wind shear biases between 200 and 850 hPa; and (g, h) potential intensity biases.

16. **Figure 16 and 17:** Again, I don't understand the tendency of the authors to try and explain everything with MJO. There are other things besides it. For instance, what about African Easterly Waves in the Atlantic?

Thank you for your suggestion. The main purpose of this paper is to introduce the FGOALS-f3 GCMs participating in the HighResMIP and to show the simulated performance of tropical cyclones in FGOALS-f3 at 100 and 25 km horizontal resolution. The MJO is just one possible reason we use to explain the simulated performance. We agree with your view that the ENSO and other large-scale factors are important in modulating tropical cyclone activities in many basins. We therefore do not discuss the physical relationship between the MJO and tropical cyclones and

have added an analysis of the ENSO (Figure 10) and large-scale factors (Figure 11). In the discussion section, we only give the results of the GPI pattern calculated using the multi-model mean (Figure 13).

17. **Tables 1-3:** There's no information presented in the paper on the length of simulations, etc.

Thank you for your correction. The length of the simulations is an important message and we have added it to Table 2.

**Table 2.** Comparison of resolutions, time steps and length of simulations in FGOALS-f3 for HighResMIP Tier 1.

| Model configuration | 100 km FGOALS-f3 | 25 km FGOALS-f3 |
|---|---|---|
| Horizontal resolution | C96 (about 100 km) | C384 (about 25 km) |
| Number of vertical layers | 32 | 32 |
| Number of vertical remapping operations per physical time step with dynamical integration (k_split) | 2 | 6 |
| Number of small dynamic time steps between the vertical remapping operations (n_split) | 6 | 15 |
| Time step of dynamical core (min) | 30 | 30 |
| Time step of physical processes (min) | 30 | 30 |
| Frequency of radiative transmission (h) | 1 | 1 |
| Minimum time step of microphysics scheme (s) | 150 | 150 |
| Length of simulations | 1950–2014 | 1950–2014 |

---

## Author Response (AR2)

**Reply to Reviewer #1**

Thank you very much for your help to make our study more meaningful. We believe the quality of the manuscript has improved significantly after considering your valuable suggestions. The following is our point-by-point reply to your comments. The revisions made in response to your comments can be followed in the modified version of the original manuscript uploaded as part of this revised submission.

**Comments:**

The authors have addressed my comments fairly well. I just have a few additional points for them to address before the manuscript can be accepted.

1) Figures 2 and 3: Include a 4th panel showing differences between the model at 25 km and observations.

   We thank you for your valuable suggestion. We have checked figure 2 and 3 in the manuscript and found the differences between the model at 25 km and observations were shown in figure 2b and 3b, which can meet your suggestion.

2) Figures 4, 5, 6, 7 and 8: Is there a reason why the South Indian Ocean basin was left out?

   Thanks for your valuable question. The simulated tropical cyclone (TC) activities in South Indian Ocean (SI) can also be given. As your suggestion, we have added the TC information in SI on Figures 4, 5, 6, 7, 8 and table 5, which will make the manuscript more complete.

3) Table 4: The numbers of TCs is underestimated in every TC basin in the low-resolution version of the model compared to the high-resolution version, as expected. However, the numbers are the same in the western Pacific. It's hard to believe that a 100 km resolution version of the model can produce nearly 25 TCs per year in the western Pacific. Can the authors explain this?

Thanks for your valuable comment. Our previous work suggests that the FGOALS-f3 model in 100km resolution condition can resolve more than 70% of TC counts globally (Li et al., 2019). The possible reason is that the model can reproduce tropical waves when developed a Resolving Convective Precipitation (RCP) scheme. This scheme involves calculating the microphysical processes in the cumulus scheme for both deep and shallow convection, which the description of sub-grid convection is weekend. A more reasonable background of tropical wave in our model will be good for TC formation, especially in the western pacific (WP). On the other hand, the dynamical core in models also affects the TC simulating performance. The atmospheric dynamical core used in FGOALS-f3 is the finite-volume cubed-sphere dynamical core, which is similar to the GFDL global atmosphere and land model AM4.0 (Zhao et al., 2018a). We can find that the GFDL model in 100km resolution version can also reproduce TC activities globally (Zhao et al., 2018b). These results indicate that the model can reproduce TC in middle resolution when the coordination between the dynamical core a physical process is handled well (Zhao et al., 2012). However, the tracker scheme is also sensitive to the simulating result (Roberts et al., 2020a, 2020b). Our next work will reduce these uncertainties when considering more tracker schemes (e.g., TempestExtremes, TRACK, and TSTORM).

Reference

Ullrich, P. A., Zarzycki, C. M., McClenny, E. E., Pinheiro, M. C., Stansfield, A. M., and Reed, K. A.: TempestExtremes v2.1: a community framework for feature detection, tracking, and analysis in large datasets, Geosci. Model Dev., 14, 5023–5048, https://doi.org/10.5194/gmd-14-5023-2021, 2021.

Hodges, Kevin, Alison Cobb, and Pier Luigi Vidale. "How well are tropical cyclones represented in reanalysis datasets?." Journal of Climate 30.14 (2017): 5243-5264.

Roberts, Malcolm John, et al. "Projected future changes in tropical cyclones using the CMIP6 HighResMIP multimodel ensemble." Geophysical research letters 47.14 (2020a): e2020GL088662.

Roberts, Malcolm John, et al. "Impact of model resolution on tropical cyclone simulation using the HighResMIP–PRIMAVERA multimodel ensemble." Journal of Climate 33.7 (2020b): 2557-2583.

Zhao, Ming, et al. "Simulations of global hurricane climatology, interannual variability, and response to global warming using a 50-km resolution GCM." Journal of Climate 22.24 (2009): 6653-6678

Zhao, Ming, et al. "The GFDL global atmosphere and land model AM4. 0/LM4. 0: 2. Model description, sensitivity studies, and tuning strategies." Journal of Advances in Modeling Earth Systems 10.3 (2018a): 735-769.

Zhao, Ming, et al. "The GFDL global atmosphere and land model AM4. 0/LM4. 0: 1. Simulation characteristics with prescribed SSTs." Journal of Advances in Modeling Earth Systems 10.3 (2018b): 691-734.

Zhao, Ming, Isaac M. Held, and Shian-Jiann Lin. "Some counterintuitive dependencies of tropical cyclone frequency on parameters in a GCM." Journal of the Atmospheric Sciences 69.7 (2012): 2272-2283.

Li, Jinxiao, et al. "Evaluation of FAMIL2 in simulating the climatology and seasonal-to-interannual variability of tropical cyclone characteristics." Journal of Advances in Modeling Earth Systems 11.4 (2019): 1117-1136.

4) Replace Balaguru et al. (BAMS, 2020) with Balaguru et al. (JAMES, 2020). The wrong publication was cited in this case.

Thanks for your correction. We have replaced Balaguru et al. (BAMS, 2020) with Balaguru et al. (JAMES, 2020).

Reference

Balaguru, K., Foltz, G. R., Leung, L., R., Kaplan, J., Xu, W., Reul, N., and Chapron, B. et al.: Pronounced impact of salinity on rapidly intensifying tropical cyclones, Bull. Am. Meteorol. Soc., 101.9, E1497-E1511, https://doi.org/10.1175/BAMS-D-19-0303.1, 2020.

Balaguru, K., Leung, L., R., Van Roekel, L., P., Golaz, J., C., Ullrich, P., A., Caldwell, P., M., Hagos, S., M., Harrop, B., E., and Mametjanov, A.: Characterizing tropical cyclones in the energy exascale earth system model Version 1, J. Adv. Model. Earth Sy., 12, e2019MS002024. https://doi.org/10.1029/2019MS002024.